# Distributed functions of prefrontal and parietal cortices during sequential categorical decisions

Yang Zhou[1,2][†]*, Matthew C Rosen[1][†], Sruthi K Swaminathan[1], Nicolas Y Masse[1], Ou Zhu[1], David J Freedman[1,3]*

[1]Department of Neurobiology, The University of Chicago, Chicago, United States; [2]School of Psychological and Cognitive Sciences, PKU-IDG/McGovern Institute for Brain Research, Peking-Tsinghua Center for Life Sciences, Peking University, Beijing, China; [3]Neuroscience Institute, The University of Chicago, Chicago, United States

**Abstract** Comparing sequential stimuli is crucial for guiding complex behaviors. To understand mechanisms underlying sequential decisions, we compared neuronal responses in the prefrontal cortex (PFC), the lateral intraparietal (LIP), and medial intraparietal (MIP) areas in monkeys trained to decide whether sequentially presented stimuli were from matching (M) or nonmatching (NM) categories. We found that PFC leads M/NM decisions, whereas LIP and MIP appear more involved in stimulus evaluation and motor planning, respectively. Compared to LIP, PFC showed greater nonlinear integration of currently visible and remembered stimuli, which correlated with the monkeys' M/NM decisions. Furthermore, multi-module recurrent networks trained on the same task exhibited key features of PFC and LIP encoding, including nonlinear integration in the PFC-like module, which was causally involved in the networks' decisions. Network analysis found that nonlinear units have stronger and more widespread connections with input, output, and within-area units, indicating putative circuit-level mechanisms for sequential decisions.

*For correspondence:
yangzhou1@pku.edu.cn (YZ);
dfreedman@uchicago.edu (DJF)

[†]These authors contributed
equally to this work

Competing interests: The
authors declare that no
competing interests exist.

Reviewing editor: Anna Wang
Roe, Zhejiang University, China

## Introduction

The ability to compare and make decisions about sequentially presented sensory stimuli is essential for generating appropriate behavioral responses to the stimuli and events in our surroundings. Such sequential decisions require incoming sensory information to be compared to information maintained in short-term memory. Although previous work has given insight into patterns of activity across several cortical areas during sequential decision tasks, particularly those based on delayed match and delayed nonmatch to sample paradigms (*Freedman and Assad, 2006*; *Freedman et al., 2001*; *Miller and Desimone, 1994*; *Miller et al., 1991*; *Wallis et al., 2001*), the mechanisms and computations underlying sequential decisions remain largely unknown. The current study examines the pattern of neuronal activity across three interconnected frontal-parietal cortical areas—prefrontal cortex (PFC), lateral intraparietal (LIP), and medial intraparietal (MIP)—during a delayed match to category (DMC) task, which requires monkeys to indicate whether sequentially presented sample and test stimuli belong to the same category. Our group showed previously that neural activity in PFC, LIP, and MIP all encode learned categories during this task (*Freedman and Assad, 2006*; *Swaminathan and Freedman, 2012*; *Swaminathan et al., 2013*), with evidence suggesting that LIP is more directly involved in categorization of visual motion compared to MIP and PFC. However, the mechanisms by which sample and test stimuli are compared in order to generate matching (M)/ nonmatching (NM) decisions are the focus of the current study.

The lateral PFC, LIP, and MIP are important processing stages for mediating decision-making and other types of cognitive task (*Andersen and Buneo, 2002*; *Andersen et al., 1997*; *Bisley and*

*Goldberg, 2010*; *Colby and Goldberg, 1999*; *de Lafuente et al., 2015*; *Ding and Gold, 2012*; *Funahashi et al., 1993*; *Gold and Shadlen, 2007*; *Gottlieb, 2007*; *Gregoriou et al., 2009*; *Huk et al., 2017*; *Kim and Shadlen, 1999*; *Miller et al., 1996*; *Moore and Armstrong, 2003*; *Padoa-Schioppa and Assad, 2006*; *Platt and Glimcher, 1999*; *Rossi-Pool et al., 2017*; *Seger and Miller, 2010*; *Shadlen and Newsome, 1996*; *Squire et al., 2013*; *Sugrue et al., 2004*; *Zhou and Freedman, 2019*; *Zhou et al., 2016*; *Zhou et al., 2018*; *Zhou et al., 2020*). Specifically, PFC neural activity has been shown to reflect the comparison of sequential stimuli during delayed match to sample tasks using either visual or vibrotactile stimuli, and such comparison-related activity correlates with monkeys' choice behavior (*Hussar and Pasternak, 2012*; *Hussar and Pasternak, 2013*; *Romo et al., 2004*; *Rossi-Pool et al., 2016*). LIP neural activity was also recently reported to represent the match status of sequential stimuli in a delayed conjunction matching task (*Ibos and Freedman, 2017*). Moreover, MIP has been shown to be involved in planning manual movements and mediating hand movements during decision-making tasks (*Cui and Andersen, 2007*; *de Lafuente et al., 2015*; *Snyder et al., 1997*). However, because these three cortical areas have not been directly compared in the same animals performing the same tasks, it is unclear how activity within these cortical areas contributes to the comparison of sequential stimuli to form M/NM decisions.

In this study, we focus on the test period of the DMC task, during which monkeys made their M/NM decisions by comparing remembered sample information with the currently visible test stimulus. We found that test-period activity in all three cortical areas was correlated with monkeys' M/NM decisions, but in different ways. M/NM selectivity in PFC appeared with a shorter latency than in LIP and MIP, suggesting a leading role for PFC in that decision process. Meanwhile, LIP showed the strongest categorical encoding of both the previously presented sample stimulus and the currently visible test stimulus. In contrast, MIP activity primarily reflected the monkeys' arm and/or hand movements used to report their decisions, rather than more abstract decision variables. Individual neurons in PFC and LIP, but not MIP, encoded both the remembered sample and currently visible test categories simultaneously during the test period. However, sample and test information in PFC was combined in a more nonlinear fashion than in LIP, and PFC neurons— which showed a greater level of nonlinear integrative encoding, correlated more closely with the monkeys' M/NM decisions compared to neurons with more linear encoding. This suggests that nonlinear integration of sample and test information in PFC is an important information processing step for generating decision-related M/NM encoding for sequential decision making.

Previous studies from our group and others have trained artificial recurrent neural networks (RNNs) on similar behavioral tasks used in experimental neurophysiological studies—an approach that has proven helpful in exploring putative circuit computations underlying cognitive tasks, generating predictions for analyses of experimental data, and potentially for enhancing capabilities of RNNs (*Engel et al., 2015*; *Masse et al., 2019*; *Yang et al., 2019*). To further understand the circuit mechanisms of M/NM computation and the roles of nonlinear integration of task variables during M/NM decisions, we trained multi-module RNNs to perform the same DMC task and analyzed activity of model units within each module of the hidden layer during task performance. The trained RNNs exhibited high levels of behavioral task performance, and the units in each of the networks' modules showed similar patterns of activity and dynamics as in neural data from posterior parietal cortex (PPC) and PFC. In particular, this included two key observations, which mirrored those from the neural data: (1) shorter-latency M/NM selectivity in the PFC module than the LIP module and (2) decision-correlated nonlinear encoding of sample and test information in the PFC module. We also found that the M/NM selectivity in the LIP module was at least partially inherited from top-down projections from the PFC module in the RNNs, potentially giving insight into the origin of M/NM encoding observed in the experimental data. Interestingly, when we compared the patterns of connectivity among RNN units, we found that nonlinear integrative units in the PFC are like 'hubs': in comparison to the other units in the PFC module, they received greater feedforward input from the LIP units, were more recurrently connected within the PFC module, and sent greater output to the two key output units. Furthermore, through causal inactivation experiments in the RNNs, we showed that the nonlinear integrative units in the PFC module were necessary for mediating the M/NM decisions. Together, our results from both neurophysiological experiments in trained monkeys, and analysis of neural activity and connection weights in trained RNNs, suggest that nonlinear integrative encoding, like that observed in PFC, functions as a key neuronal substrate for mediating sequential decisions.

# Results

## Task and behavioral performance

Two monkeys performed a DMC task, in which they needed to (1) categorize the sample stimulus, (2) remember it during the delay period, (3) categorize the following test stimulus, (4) determine whether the test was a categorical match to the sample, and (5) report that decision by releasing or holding a manual lever (*Figure 1A*). Stimuli consisted of six random-dot motion directions that were grouped into two arbitrary learned categories, with three motion directions per category (*Figure 1B*). This corresponds to a total of 36 possible sample-test direction combinations and four sample-test category combinations. Both monkeys performed the DMC task with high accuracy (monkey A: 91%; monkey B: 97%), and greater than 80% for all 36 stimulus conditions during recordings from each of the three areas (*Figure 1C, D*, monkey A: PFC = 93% ± 4.6%, LIP = 93% ± 4.8%, MIP = 88% ± 6.3%; monkey B: PFC = 98% ± 1.3%, LIP = 96% ± 4.1%, MIP = 98% ± 2.5%). Slightly higher error rates were observed on match compared to nonmatch trials for all three datasets of

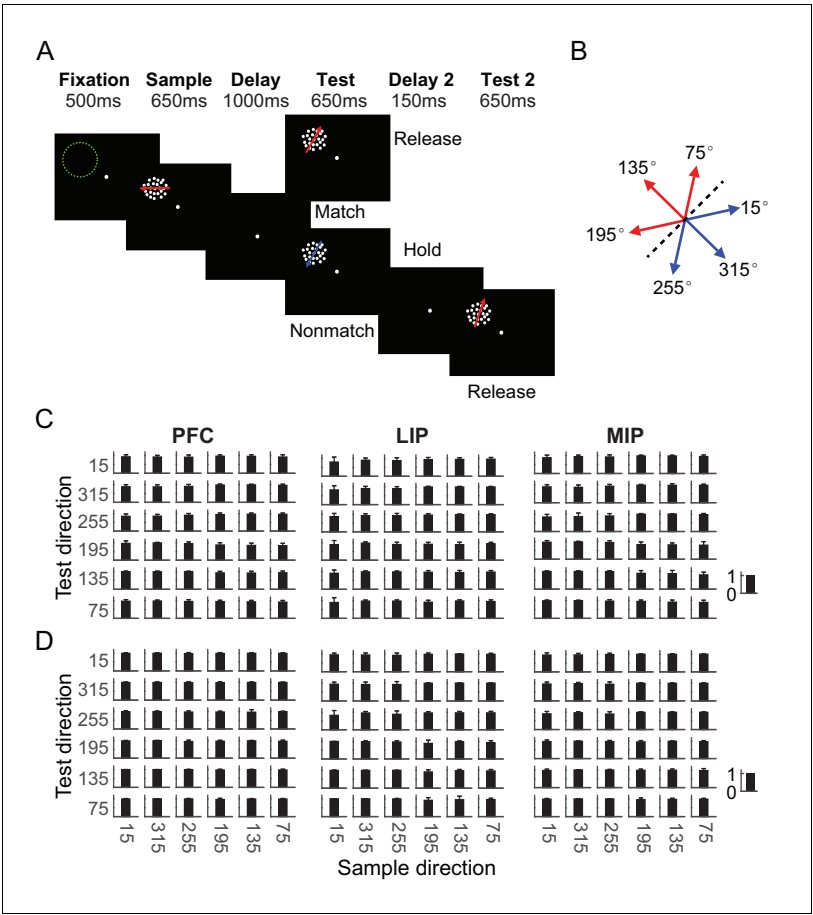

**Figure 1.** Task and behavioral performance. (**A**) Sequence of the delayed match to category (DMC) task. Monkeys needed to release a touch-bar when the categories of sample and test stimuli matched, or hold the bar and wait for the second test stimulus when they did not match. The matching (M)/nonmatching (NM) decision was required to be made during the test period to receive the reward. Stimulus offset occurred immediately after monkeys released the touched-bar. The green dashed circle indicates the position of a neuron's receptive field. (**B**) Monkeys needed to group six motion directions into two categories (corresponding to the red and blue arrows) separated by a learned category boundary (black dashed line). (**C, D**) Two monkeys' average performance (accuracy) for all stimulus conditions during recordings from prefrontal cortex (PFC), lateral intraparietal (LIP), and medial intraparietal (MIP) recordings are shown separately. Each row corresponds to one monkey. The error bar denotes standard deviation.

both monkeys (*Supplementary file 1*). These results indicate that monkeys reliably based their M/NM decisions on the category membership of both sample and test stimuli.

## M/NM selectivity in PFC, LIP, and MIP

We recorded neuronal spiking activity and local field potential (LFP) signals from PFC, LIP, and MIP in separate sessions; in each session, one cortical area was targeted while the monkey performed the DMC task (except for a subset of simultaneous PFC-LIP sessions). While neural data from the same sessions were presented in previous reports from our lab (*Swaminathan and Freedman, 2012*; *Swaminathan et al., 2013*), the current analysis focuses on decision signals in the test period of the task, which was not a primary focus of our previous work. Furthermore, our previous reports compared only pairs of cortical areas rather than all three areas, as in the current study. We focus our analysis on neurons that showed firing rates above an arbitrary threshold (the maximum of the condition averaged firing rate across the task period $\geq$ 5 spike/s) and were significantly modulated by task variables (see Materials and methods). In order to capture the neuronal correlates of the M/NM decision from sensory processing to motor planning, we analyzed neuronal activity from 50 to 350 ms following test stimulus onset since monkeys' manual responses (indicating their M/NM decisions) occurred within this epoch on 95.6% of match trials (monkey A: 246.2 ± 14.0 ms, monkey B: 282.7 ± 16.3 ms). Note that most of our subsequent analyses focus on a shorter-duration time window, ending prior to the monkeys' mean reaction time. In this time period, a substantial fraction of neurons in each cortical area (PFC: 104/145, LIP: 35/53, MIP: 60/66) showed significantly different test-period activity between match and nonmatch trials (match vs. nonmatch; p<0.01, one-way ANOVA). We classified these neurons as 'match-preferring' and 'nomatch-preferring' based on whether they showed significantly higher firing rates for match or nonmatch trials, respectively (example neurons in *Figure 2—figure supplement 1* and population activity in *Figure 2A–C*).

To characterize the time course of M/NM selectivity, we performed a receiver-operating characteristic (ROC) analysis comparing each neuron's firing rates on match and nonmatch trials using a 50 ms window advanced in 5 ms steps (*Figure 2D–F*). Both match- and nonmatch-preferring neurons showed M/NM selectivity shortly after test onset in all three areas, though the fractions of neurons preferring match or nonmatch differed among the areas. PFC showed a relatively balanced distribution of match- and nonmatch-preferring neurons (match:nonmatch, 43:61), while LIP and MIP were more biased toward match-preferringneurons (match:nonmatch, LIP: 22:13, MIP: 51:9, $P_{(PFCvs.LIP)}$ = 0.027, $P_{(PFCvs.MIP)}$ = $5.2 \times 10^{-8}$, chi-square test). In all three areas, match-preferring neurons exhibited significantly greater activity than nonmatch-preferring neurons during the test period ($P_{(PFC)}$ = 0.0372; $P_{(MIP)}$ = 0.0427; $P_{(LIP)}$ = 0.012; Wilcoxon test).

To compare the strength of M/NM selectivity between match- and nonmatch-preferring neurons, we calculated the unbiased fraction of explained variance (FEV) by the M/NM choice (see Materials and methods). On average, match-preferring neurons showed significantly greater M/NM selectivity in all three areas ($P_{(PFC)}$ = $2.6 \times 10^{-4}$; $P_{(MIP)}$ = 0.024; $P_{(LIP)}$ = 0.039; Wilcoxon test, *Figure 2—figure supplement 2*), as well as significantly shorter latency of M/NM selectivity than nonmatch-preferring neurons in PFC (118.0 vs. 128.5 ms, p=0.0071, Wilcoxon test; see Materials and methods). Differences of M/NM selectivity remained statistically significant in PFC and MIP (in both magnitude and latency), but showed only a nonsignificant trend in LIP, after equating the mean firing rates between groups of neurons (see Materials and methods, $P_{(PFC, magnitude)}$ = 0.0015; $P_{(PFC, latency)}$ = 0.0251; $P_{(LIP)}$ = 0.12; $P_{(MIP)}$ = 0.040; Wilcoxon test). This suggests that the stronger M/NM selectivity of match-preferring neurons observed in these areas is unlikely to be explained by the higher firing rates of match-preferring neurons.

## Comparing the roles of PFC, LIP, and MIP in M/NM decisions

To elucidate the relative contributions of PFC, LIP, and MIP to M/NM decisions, we first compared the time course of M/NM selectivity across areas. Since match-preferring neurons showed earlier and stronger M/NM selectivity than nonmatch-preferring neurons, and the distributions of the two groups of neurons were different among areas, we compared the M/NM selectivity of match-preferring and nonmatch-preferring neurons separately in each cortical area. Using an unbiased FEV analysis, we found that match-preferring neurons exhibited significantly shorter-latency (see Materials and methods) M/NM selectivity in PFC than in LIP and MIP (*Figure 3A*, PFC: n = 40, LIP: n = 22, MIP:

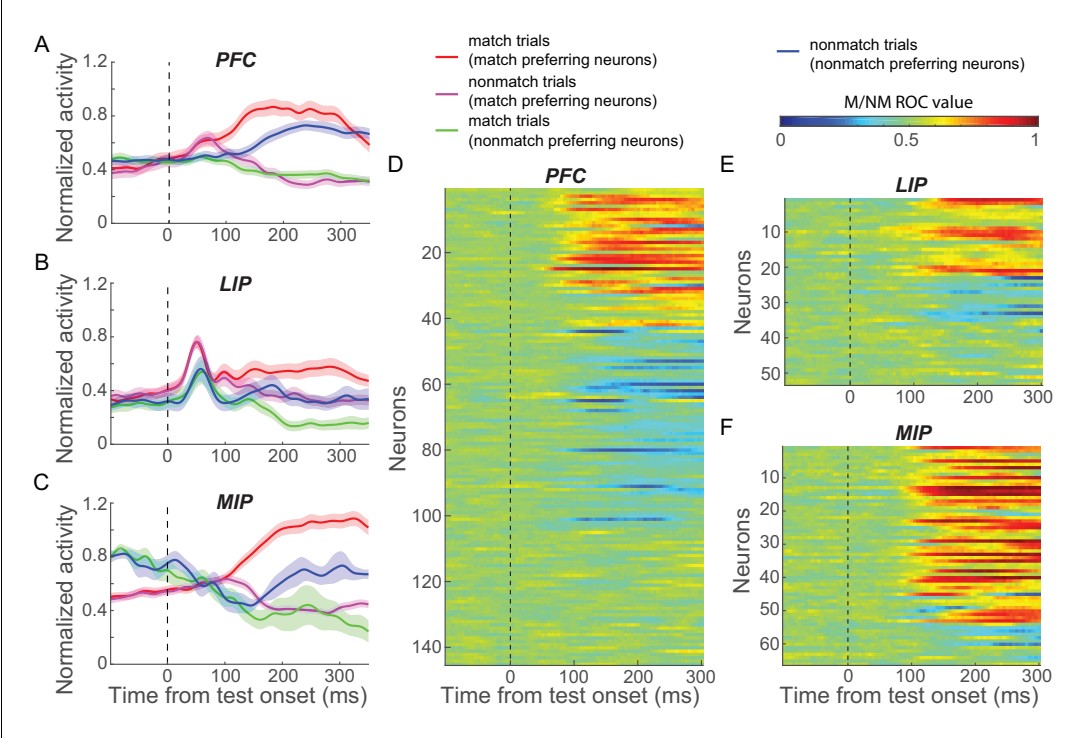

**Figure 2.** Matching (M)/nonmatching (M/NM) selectivity in prefrontal cortex (PFC), lateral intraparietal (LIP), and medial intraparietal (MIP) areas. (A–C) The normalized population activity of both match-preferring and nonmatch-preferring neurons in PFC (A), LIP (B), and MIP (C). The shaded area represents ± SEM. (D–F) The strength of the M/NM selectivity was evaluated using receiver-operating characteristic analysis for all neurons in PFC (D), LIP (E), and MIP (F). Values close to 0.0 and 1.0 correspond to strong encoding preference for nonmatch and match, respectively. Values of 0.5 indicate no M/NM selectivity.

The online version of this article includes the following figure supplement(s) for figure 2:

**Figure supplement 1.** Examples of match and nonmatch-preferring neurons in prefrontal cortex (PFC), lateral intraparietal (LIP), and medial intraparietal (MIP) areas.

**Figure supplement 2.** Comparison of the matching/nonmatching (M/NM) selectivity between match-preferring and nonmatch-preferring neurons.

n = 50; $P_{(PFC\ vs.\ LIP)}$ = 0.026, $P_{(PFC\ vs.\ MIP)}$ = 0.0058, Wilcoxon test). Nonmatch-preferring neurons showed a similar trend (*Figure 3B*), although the difference did not rise to significance—likely due to the small population of nonmatch-preferring neurons in LIP and MIP (PFC: n = 50, LIP: n = 12, MIP: n = 9; $P_{(PFC\ vs.\ LIP)}$ = 0.11, $P_{(PFC\ vs.\ MIP)}$ = 0.13, Wilcoxon test). We also examined the M/NM selectivity of the LFP signal, which likely reflects the activity and/or computations within the local network (*Burns et al., 2010*; *Logothetis et al., 2001*). The mean amplitude of the LFP in PFC also showed significantly shorter-latency M/NM selectivity than in LIP and MIP (*Figure 3C*, $P_{(PFC\ vs.\ LIP)}$ = $8.9 \times 10^{-4}$, $P_{(PFC\ vs.\ MIP)}$ = 0.014, Wilcoxon test). We determined that the shorter-latency M/NM selectivity in PFC compared to PPC was not due to differences in latency on sessions that targeted each brain area, as we observed similar results in a different dataset from another study (conducted in different monkeys) using the DMC task in our lab (*Masse et al., 2017*), in which neuronal activity was recorded simultaneously from PFC and PPC using a semi-chronic multielectrode approach (see Materials and methods). As shown in *Figure 3—figure supplement 1*, PFC neurons showed significantly shorter-latency M/NM selectivity than PPC neurons in that study (p=0.0158, Wilcoxon test). Furthermore, the raw LFP amplitude, which was recorded simultaneously from the two areas, showed shorter-latency M/NM selectivity in PFC than in PPC in 52 of 58 recording sessions from both monkeys (PFC:PPC, monkey Q: 151.2 ms:199.1 ms, monkey W: 157.3 ms:171.7 ms, the difference in 46 sessions reached statistical significance p<0.05, Wilcoxon test; Figure S3D–G). Together, the shorter latency M/NM selectivity in PFC compared with PPC is consistent with a preferential role for PFC in M/NM decisions.

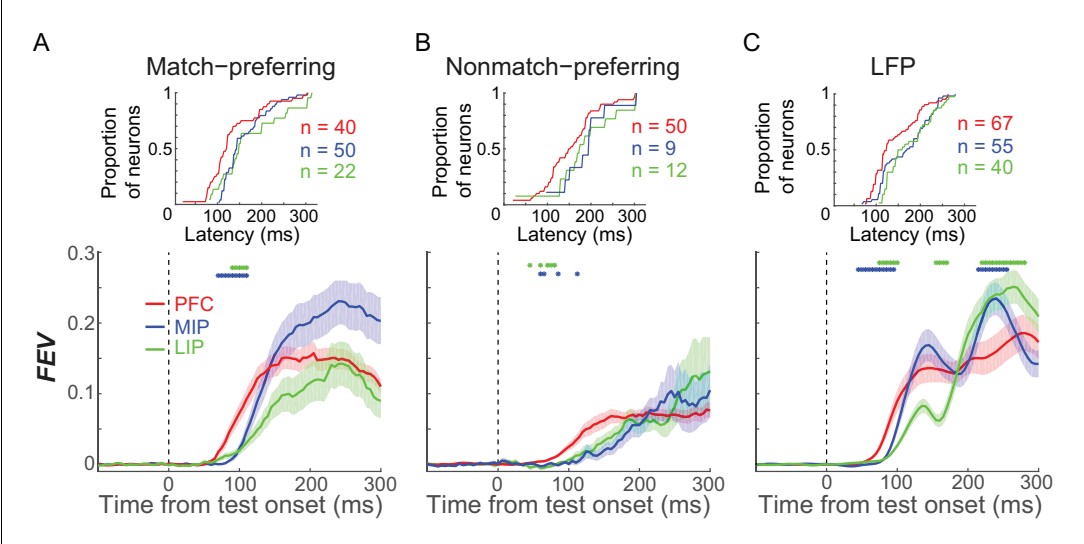

**Figure 3.** The comparison of matching/nonmatching (M/NM) selectivity between prefrontal cortex (PFC), lateral intraparietal (LIP), and medial intraparietal (MIP) areas. (A, B) The magnitude and time course of M/NM selectivity was determined using unbiased fraction of explained variance (FEV). Different colors represent different cortical areas, and the shaded area represents ± SEM. The blue dots denote the time points for which there were significant differences between PFC and MIP, while the green dots denote the time points for which there were significant differences between PFC and LIP (p<0.05, Wilcoxon test). The upper inset figures show the cumulative distribution of the latency of M/NM selectivity. (C) The M/NM selectivity of LFP amplitude in PFC, LIP, and MIP, which is shown in the same format as (A). The LFP signal from all recording channels in each area is included.

The online version of this article includes the following figure supplement(s) for figure 3:

**Figure supplement 1.** Comparison of matching/nonmatching (M/NM) selectivity between prefrontal cortex (PFC) and PPC in the delayed match to category (DMC) task.

Second, we tested whether the M/NM selectivity observed in each cortical area correlated with specific cognitive processes, such as the comparison of sample and test categories, or movement planning/initiation. For this purpose, we compared the activity of match-preferring neurons between match and nonmatch trials aligned to the monkeys' lever release. In the DMC task, monkeys released the lever during both the first test period of match trials and the second test period of non-match trials. However, the decision about the match status of the test stimulus occurred during the first test period of both types of trials since the second test stimulus was only shown on nonmatch trials and was always a match (requiring a lever release). Match-preferring neurons in PFC showed greater activity preceding and up to the time of the lever release on match trials compared to non-match trials (200–0 ms prior to the hand movement, *Figure 4A*, p=0.0018, paired t-test). In LIP, match-preferring neurons showed greater activity during match trials vs. nonmatch trials only prior to, but not coincident with, the hand movement (200–100 ms prior to the hand movement, *Figure 4B*, p=0.0024; −100–0 ms, p=0.21; paired t-test). The activity of match-preferring neurons in MIP during both match and nonmatch trials was very similar both before and during the hand movement (200–0 ms prior to the hand movement, *Figure 4C*, p=0.99, paired t-test). These results suggest that match-preferring neurons in PFC and LIP are more involved in nonmotor functions during M/NM decisions, such as the comparison of sample and test categories, while match-preferring neurons in MIP are primarily involved in motor functions such as planning and/or initiating hand/arm movements.

Furthermore, we examined how neuronal M/NM selectivity covaried with the monkeys' reaction time (RT) across the three cortical areas. To examine this, we separated match trials into two equal-sized RT subgroups (fast and slow) for each neuron (fast:slow: LIP = 240.6:287.3 ms, MIP = 243.9:293.1 ms, PFC = 233.4:278.3 ms, see Materials and methods). We then compared M/NM selectivity for these subgroups in each area. Both the unbiased FEV and the SVM analyses revealed significantly shorter-latency M/NM selectivity in MIP for faster vs. slower RT trials (*Figure 4F* and *Figure 4—figure supplement 1C*; $P_{(FEV)} = 0.0011$, paired t-test; $P_{(SVM)} < 0.02$, bootstrap), but not in

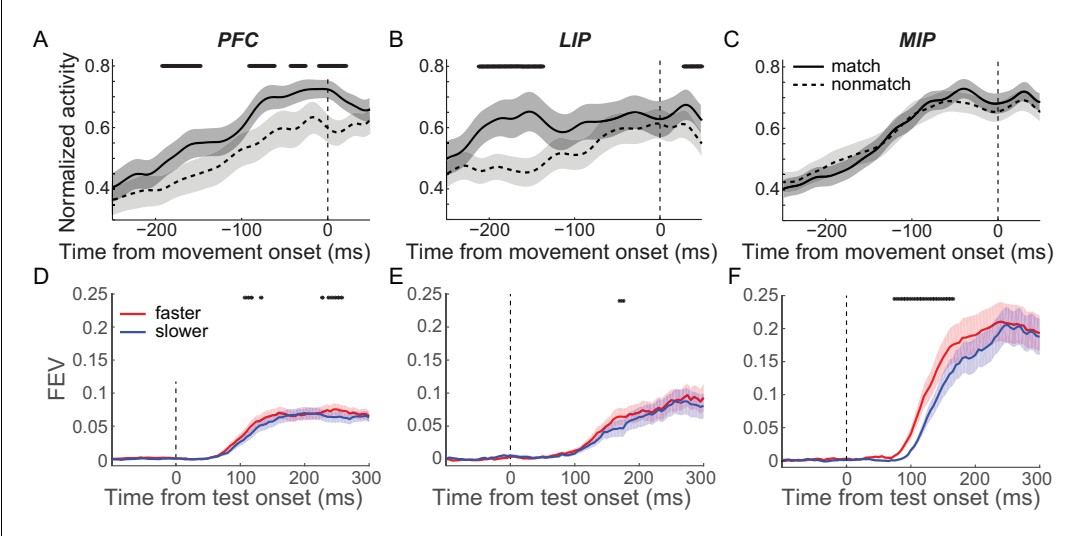

**Figure 4.** Matching/nonmatching (M/NM) selectivity in medial intraparietal (MIP) cortex but not prefrontal cortex (PFC) and lateral intraparietal (LIP) cortex primarily correlated with monkeys' hand movement for reporting M/NM decisions. (**A–C**) The population activity of match-preferring neurons during match and nonmatch trials when activity was aligned to the start of hand movement. The black stars mark the time periods for which there were significant differences (p<0.01, paired t test). (**D–F**) The time course and magnitude of the M/NM selectivity in faster trials (red) and slower trials (blue) were evaluated using unbiased fraction of explained variances (FEVs) for PFC (**D**), LIP (**E**), and MIP (**F**). The shaded area represents ± SEM, and the black dots denote the time points for which there were significant differences (p<0.01, paired t-test).

The online version of this article includes the following figure supplement(s) for figure 4:

**Figure supplement 1.** Matching/nonmatching (M/NM) selectivity in medial intraparietal (MIP) cortex correlated with monkeys' reaction times (RTs).

PFC or LIP (*Figure 4D, E* and *Figure 4—figure supplement 1A, B*; PFC: $P_{(FEV)}$ = 0.20, paired t-test; $P_{(SVM)}$ > 0.3, bootstrap; LIP: $P_{(FEV)}$ = 0.60, paired t-test; $P_{(SVM)}$ > 0.40, bootstrap). Given the longer latency of M/NM selectivity in MIP and the preponderance of neurons preferring match conditions (which were accompanied by arm/hand movement), these results further suggest that M/NM selectivity in MIP is primarily associated with motor aspects of the monkeys' M/NM decisions.

## Integrative sample and test category representations in PFC and LIP

To solve the DMC task, the category membership of both sample and test stimuli needs to be compared or integrated to form the M/NM decision. To gain insight into the basis for this integration, we examined how test-period activity in the three cortical areas encoded the previously presented sample category and the currently visible test category, and how sample and test category representation is related to the M/NM decision process. We first quantified the neuronal representation of sample and test categories in each area during the first test period using a two-way ANOVA on test-period activity with sample and test categories as factors. The magnitude of the category selectivity was quantified using unbiased fraction explained variance (see Materials and methods). We focused on the time window preceding the mean RTs of both monkeys (0–250 ms after test onset) since sample and test category information must be integrated before the monkeys' M/NM choice. Test-period activity in LIP showed significantly stronger encoding of both sample and test categories than PFC and MIP (*Figure 5—figure supplement 1*, $P_{(LIP\ vs.\ PFC,\ sample)}$ = 0.0083, $P_{(LIP\ vs.\ MIP,\ sample)}$ = 0.00076, $P_{(LIP\ vs.\ PFC,\ test)}$ = 0.0094, $P_{(LIP\ vs.\ MIP,\ test)}$ = $8.3 \times 10^{-6}$, Wilcoxon test)—consistent with our previous findings (*Swaminathan and Freedman, 2012*; *Swaminathan et al., 2013*)—suggesting that LIP is more directly involved in category computation than PFC. Meanwhile, test-period activity in both PFC and LIP both showed a combined encoding of sample and test category information, but this was not observed in MIP (*Figure 5A–C*, PFC: $FEV_{(sam)}$ = 0.0399, p=$6.7 \times 10^{-8}$, $FEV_{(tes)}$ = 0.0108, p=$1.7 \times 10^{-5}$; LIP: $FEV_{(sam)}$ = 0.1143, p=$3.5 \times 10^{-6}$, $FEV_{(tes)}$ = 0.0402, p=$5.7 \times 10^{-4}$; MIP: $FEV_{(sam)}$ = 0.0230, p=$3.5 \times 10^{-5}$, $FEV_{(tes)}$ = 0.0026, p=0.1146; paired t-test). This raises a question about the manner by which sample and test information is simultaneously encoded in both PFC and

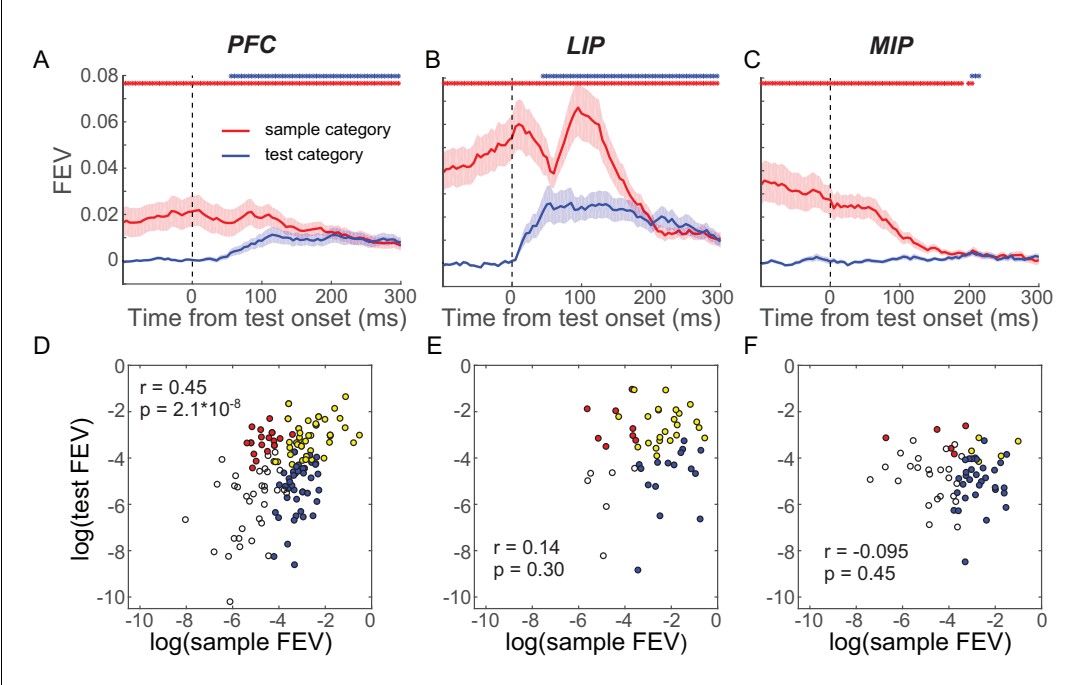

**Figure 5.** Sample and test category representation in prefrontal cortex (PFC), lateral intraparietal (LIP), and medial intraparietal (MIP) areas. (A–C) The selectivity of sample category (red) and test category (blue) was evaluated using the unbiased fraction of explained variance (FEV) for all neurons in PFC (A), LIP (B), and MIP (C). The shaded area represents ± SEM. The red and blue dots represent the time points for which the sample and test category selectivity are significantly greater than chance level (p<0.01, paired t-test), respectively. (D–F) The correlations between sample category and test category selectivity (using FEV) during the test period for all neurons in PFC (D), LIP (E), and MIP (F). Each symbol represents a single neuron. Yellow dots denote neurons that showed significantly mixed sample test category selectivity. The blue and red dots denote the neurons that showed only significant sample category or test category selectivity, respectively, and the black circles denote the neurons that did not show significant category selectivity (one-way ANOVA test, p<0.01).

The online version of this article includes the following figure supplement(s) for figure 5:

**Figure supplement 1.** Comparisons of category selectivity among prefrontal cortex (PFC), lateral intraparietal (LIP), and medial intraparietal (MIP) areas.

LIP—specifically whether sample and test information are encoded independently (e.g., additively), reflecting linear-like integration. Alternatively, sample and test information might be combined in a more nonlinear fashion in either or both areas, reflecting local computation.

To better understand how PFC and LIP integrate sample and test category information, we examined how each area simultaneously encoded the sample and test categories during the test period. First, we asked whether there was a correlation between the strength of neurons' sample and test category encoding. A positive correlation would suggest that a special pool of neurons is preferentially involved in encoding both the remembered sample and currently visible test categories, indicative of sample-test category integration at the single-neuron level. In contrast, a negative correlation or zero correlation would indicate that there is less overlap of the neurons' sample and test category encoding. We calculated the unbiased FEV of sample and test category encoding for each neuron's test-period activity and found PFC and LIP neurons that showed both sample and test category selectivity. However, the correlation between sample and test category representations at the population level differed between PFC and LIP. In PFC, there was a significant positive correlation between sample and test category selectivity shortly after test onset (*Figure 5D*, r = 0.38, p=2.9 × 10$^{-6}$, t-test), while in LIP, they were not correlated (*Figure 5E*, r = 0.086, p=0.54, t-test). The positive correlation in PFC was also evident by using the category tuning index (rCTI) to quantify category selectivity (r = 0.33, p=0.001) (see Materials and methods). Furthermore, neuronal sample category selectivity prior to test onset in PFC but not LIP (−50–50 ms relative to test onset) was correlated with M/NM selectivity during the test period (100–200 ms after test onset) (PFC: r = 0.33, p=6.1 × 10$^{-5}$; LIP: r = −0.20, p=0.15), suggesting that PFC neurons with greater sample category encoding before test onset were more likely to be involved in the M/NM computation. These results

suggest that single neurons in PFC, but not in LIP, integrate sample and test category information. MIP is unlikely to be involved in such an integration process as very few MIP neurons showed significant encoding of test category (*Figure 5F*).

Next, we examined whether sample and test category selectivity was integrated in a linear or nonlinear manner in PFC and LIP. In the DMC task, there were four possible combinations of sample and test categories (i.e., $S_1T_1$, where both sample and test stimuli are category 1, $S_1T_2$, $S_2T_1$, and $S_2T_2$). Examining the correlation of test category selectivity between the two sample category conditions ($S_1T_1$ vs. $S_1T_2$ and $S_2T_1$ vs. $S_2T_2$) provides insight into the way in which sample and test categories are integrated. We attempt to differentiate between two possible outcomes, with each suggestive of a particular kind of representation: (1) test category selectivity that added linearly to the existing sample category selectivity, in which the test category selectivity in the two sample category conditions would be similar in sign and magnitude (positive correlation); and (2) test category selectivity that combined nonlinearly with sample selectivity, in which the test category selectivity for the two sample category conditions would be different (negative or no correlation). In particular, a negative correlation would result if the neuron shows an opposite test category preference between the two sample category conditions (i.e., M/NM selectivity). We evaluated these possibilities in PFC and LIP using two approaches. The first was an ROC analysis, which quantified, for each neuron, test category selectivity in each of the two sample category conditions, and then calculated their correlation across the population in each area. This revealed a positive correlation in LIP shortly following test onset (*Figure 6B* maximum r = 0.7052, p<0.001, t-test), but not much in PFC (*Figure 6A* maximum r = 0.2472, p=0.0027). Second, we trained a decoder (SVM) to classify test category using the trials in one sample category condition (e.g., $S_1T_1$ and $S_1T_2$), and tested the decoder using trials from the other sample category condition (e.g., $S_2T_1$ and $S_2T_2$). Decoder performance is expected to be above chance if the test category selectivity was similar in each of the two sample category conditions or below chance if the test category selectivity differed based on which sample stimulus has been shown on that trial. As shown in *Figure 6C, D*, decoder accuracy is significantly greater than chance shortly after test onset in LIP (maximum accuracy = 0.7738, p<0.01, bootstrap), but does not

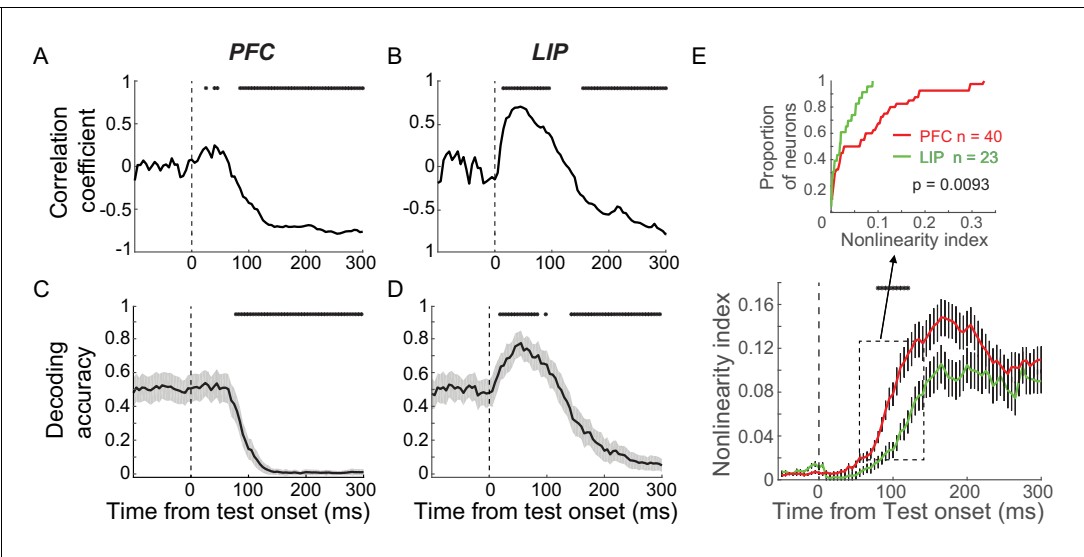

**Figure 6.** Mixed category selectivity was more nonlinear in prefrontal cortex (PFC) than in lateral intraparietal (LIP) cortex. (**A, B**) The correlation coefficient between the test category selectivity (using receiver-operating characteristic [ROC] value) of two sample category conditions is shown for both PFC (**A**) and LIP (**B**). The black dots mark the time points for which the correlation was statistically significant (p<0.01, t-test). (**C, D**) The decoding performance of a test category classifier using neuronal activity in PFC (**C**) and LIP (**D**). The support vector machine (SVM) classifier was trained by using activity from one sample category condition (e.g., $S_1T_1$ vs. $S_1T_2$) and tested with activity from the other sample category conditions (e.g., $S_2T_1$ vs. $S_2T_2$). The shaded area represents ± STD, and the black stars mark the time points for which the decoder performance is significantly different from chance level (bootstrap, p<0.05). (**E**) The nonlinearity index of mixed category-selective neurons in both PFC (red) and LIP (green). The shaded area represents ± SEM, and the black stars denote the time points for which there is a significant difference between LIP and PFC (unpaired t-test, p<0.05). The upper panel shows the cumulative distribution of nonlinearity index shortly after test onset (50–150 ms after test onset) for both PFC and LIP neurons.

exceed chance in PFC (maximum accuracy = 0.5381, p>0.50, bootstrap). Together, these results suggest that the combination of sample and test category information during the test period is more consistent with linear integration in LIP compared to PFC.

We quantified the degree to which the integrated sample-test category encoding during the test period was linear or nonlinear by calculating a nonlinearity index for LIP and PFC neurons, which showed both sample and test category selectivity in both LIP and PFC. The nonlinearity index was defined as the absolute difference between the test category selectivity in two sample category conditions quantified by the FEV between $S_1T_1$ and $S_1T_2$, or $S_2T_1$ and $S_2T_2$ (see Materials and methods). The value of the nonlinearity index, which is not expected to be affected by linearly combined category selectivity or M/NM selectivity, can range from 0 to 1. Values near 0 indicate linear-like combined encoding of the two factors, while increasing values indicate nonlinear combination of sample and test category selectivity. Because the neuronal activity shortly before monkeys' M/NM choice mainly correlated with monkeys' M/NM choice (*Figure 6*), we focus on the time window shortly after test onset (50–150 ms). As shown in *Figure 6E*, the nonlinearity index is significantly greater in PFC than LIP during the early test period (p=0.0093, unpaired t-test). This suggests greater nonlinearity in the combination of sample and test category representation in PFC compared to LIP, independent of the strength of M/NM selectivity in these two areas. The more linearly integrated sample and test category representations in LIP suggest that it is better suited for independently encoding the remembered sample stimulus and currently visible test stimulus, perhaps facilitating readout of these variables by downstream cortical areas. Considering the observation of shorter-latency M/NM selectivity in PFC than LIP and MIP (*Figure 3*), the positively correlated and more nonlinear integration of sample-test category selectivity in PFC is consistent with it combining remembered sample and visible test category information in order to facilitate M/NM decisions.

## Nonlinear PFC encoding was preferentially engaged in M/NM decisions

The results presented so far suggest that PFC is more involved in integrating sample and test category information to form M/NM decisions compared to LIP and MIP. We tested this idea more directly by assessing the relationship between PFC neurons' activity and monkeys' M/NM decisions as a function of the linearity or nonlinearity of their selectivity for sample and test categories. As in a previous study (*Lindsay et al., 2017*), we performed a two-way ANOVA on test-period activity (0–250 ms after test onset) with sample and test categories as factors to quantify the selectivity profile of PFC neurons (*Figure 7—figure supplement 1*). This allowed us to identify two populations of PFC neurons: (1) linearly integrating neurons (LIN), which exhibited main effects of both sample and test categories (p<0.01) but a nonsignificant interaction term; and (2) nonlinearly integrating neurons (NIN), which exhibited main effects of both sample and test categories (p<0.01), as well as a significant interaction term (p<0.01). We also identified nonmixed-selective neurons (NMN), which showed a significant effect of only sample category, test category or their interaction, but not mixed sample and test category encoding. Note that the M/NM-selective NMNs are different from the NINs as they did not show combined sample and test category encoding. *Figure 7—figure supplement 2* shows test-period activity of three NINs, each of which encoded both sample and test categories, and preferentially responded to one of the four sample-test category combinations. In order to test whether the NINs were more involved in mediating DMC task performance than the other groups of PFC neurons, we first compared the strength with which sample and test category information was encoded among the NINs, LINs, and NMNs using the unbiased FEV. Interestingly, NINs showed significantly stronger sample category encoding than either the LINs or sample category-selective NMNs in PFC (*Figure 7A*, $P_{(NIN\ vs.\ LIN)}$ = 0.0036; $p_{(NIN\ vs.\ NMN)}$ = $4.6 \times 10^{-8}$; Wilcoxon test), while the strength of sample category selectivity did not differ between LINs and sample category-selective NMNs (p=0.0917, Wilcoxon test). We also quantified the relationship between the nonlinear integrative encoding and category representation of neurons in both PFC and LIP. This reveled a positive correlation between the degree of nonlinear integrative encoding and the magnitude of sample category selectivity in PFC but not LIP (*Figure 7B* and *Figure 7—figure supplement 3A*, PFC: r = 0.48, p=$1.4 \times 10^{-9}$; LIP: r = 0.18, p=0.19), suggesting that sample category encoding in PFC but not LIP was primarily mediated by NINs. However, NINs did not show significantly greater test category selectivity than LINs and test category-selective NMNs ($p_{(NIN\ vs.\ LIN)}$ = 0.4642, $p_{(NIN\ vs.\ NMN)}$ = 0.2235). This might be because PFC showed weaker test category selectivity compared to LIP and

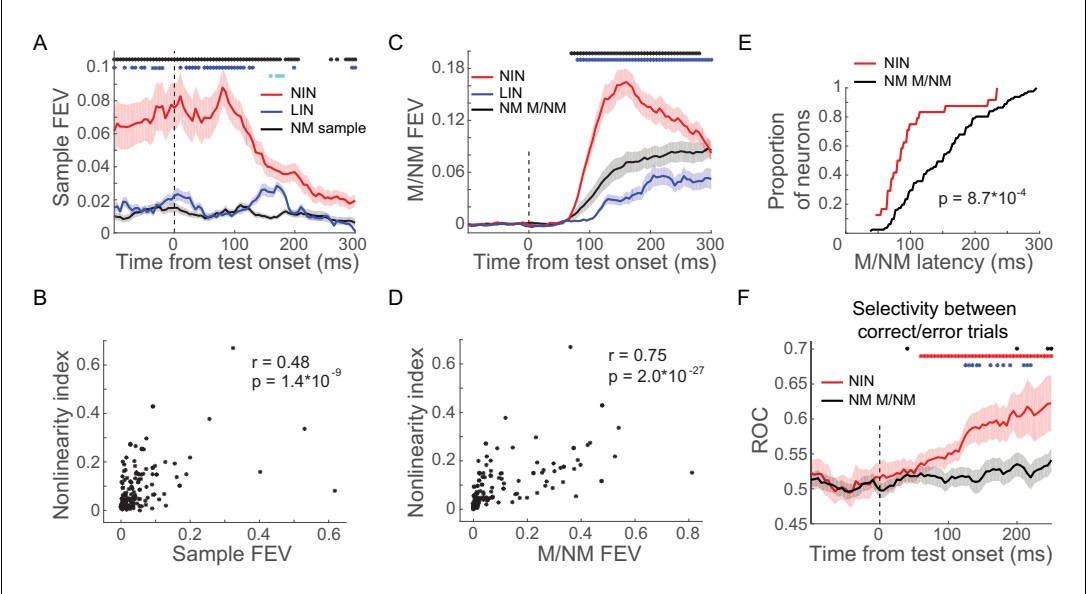

**Figure 7.** Nonlinearly integrating neurons (NIN) in prefrontal cortex (PFC) were more engaged in the delayed match to category (DMC) task. (**A**) The sample category selectivity of the NIN, linearly integrating neurons (LIN), and nonmixed sample category-selective neurons in PFC were compared using fraction of explained variance (FEV). The shaded area denotes ± SEM. The blue and black dots denote the time point for which the NINs were significantly different from LINs and nonmixed sample category-selective neurons (NM sample), respectively; while the cyan dots denote the time point for which there was significant difference between LIN and NM sample (p<0.05, Wilcoxon test). (**B**) The correlation between sample category selectivity and nonlinearity indices of PFC neurons. Each dot denotes one single neuron. (**C**) The matching/nonmatching (M/NM) selectivity of the NINs, LINs, and the nonmixed M/NM-selective neurons (NM M/NM) in PFC were compared using FEV. The colored dots denote the statistical significance in the same format as in (**A**). (**D**) Correlation between M/NM selectivity and nonlinearity index of PFC neurons. (**E**) The cumulative distribution of the latency of M/NM selectivity for NINs and NM M/NM. (**F**) The change in activity on incorrect match trials relative to correct match trials was evaluated using receiver-operating characteristic (ROC) for both NINs and NM M/NM neurons. The shaded area denotes ± SEM. The red and black dots denote the time points for which the activity changes of NINs and NM M/NM neurons were statistically significant (p<0.05, paired t-test), respectively; while the blue dots denote the time points for which there were significant differences between NINs and the NM M/NM neurons (p<0.05, Wilcoxon test).

The online version of this article includes the following figure supplement(s) for figure 7:

**Figure supplement 1.** Selectivity profile of prefrontal cortex (PFC) (**A**) and lateral intraparietal (LIP) (**B**) neurons.

**Figure supplement 2.** Nonlinearly integrating neurons (NIN) in prefrontal cortex (PFC).

**Figure supplement 3.** Correlation between nonlinearly integrative encoding and encoding of task variables in lateral intraparietal (LIP) cortex.

may therefore be less involved in rapidly encoding the currently visible test category compared to LIP.

Furthermore, NINs showed significantly greater and shorter-latency M/NM selectivity than the NMNs in PFC (*Figure 7C, E*, p(latency) = $8.7 \times 10^{-4}$, p(magnitude) = $5.6 \times 10^{-4}$, Wilcoxon test; focusing on M/NM-selective neurons in both groups). We did not include LIN for this analysis since they did not show significant M/NM selectivity based on our criteria. To better clarify this, we calculated the correlation between the degree of nonlinear integrative encoding and the M/NM selectivity of PFC neurons (both magnitude and latency). We found a significant positive correlation between the degree of nonlinear integrative encoding and the magnitude of M/NM encoding (r = 0.75, p=$2.0 \times 10^{-27}$), as well as a significant negative correlation between the degree of nonlinear integrative encoding and the latency of M/NM selectivity of PFC neurons (r = $-0.44$, p=$2.3 \times 10^{-5}$). These correlations suggest that the PFC neurons showing greater nonlinear category encoding played a preferential role in M/NM computation. To ensure that this difference between NINs and other PFC neurons was not due to differences in firing rates among the different groups, we compared the mean test-period activity among these groups of neurons and did not find significant differences (NIN: 12.8 spike/s; LIN: 11.0 spike/s, $p_{(NIN \ vs. \ LIN)}$ = 0.9670; category-selective NMN: 12.2 spike/s, $p_{(NIN \ vs. \ NMN)}$ = 0.7730; M/NM-selective NMN: 11.9 spike/s, $p_{(NIN \ vs. \ NMN)}$ = 0.7529, Wilcoxon test).

We also tested whether the activity of NINs was more closely correlated with the monkeys' M/NM decisions compared to M/NM-selective NMNs in PFC. To do so, we compared neuronal activity on correct and incorrect match trials (in which monkeys should have released the lever in response to the first test stimulus; monkeys made very few errors on nonmatch trials) with an ROC analysis (see Materials and methods). ROC values greater than 0.5 indicate that a neuron's M/NM selectivity covaries with the monkey's trial-by-trial M/NM choices, while values near or lower than 0.5 indicate no correlation or anti-correlation, respectively. As shown in *Figure 7F*, significantly elevated ROC values indicate that the activity of NINs (but not M/NM-selective NMNs) reflected the monkeys' trial-by-trial M/NM decisions (NIN: ROC = 0.604, p=0.0019; PN: ROC = 0.529, p=0.4325, paired t-test). Furthermore, the difference in NIN activity between correct and incorrect match trials (measured by ROC) was greater than that of M/NM-selective NMNs (p=0.0178, Wilcoxon test), suggesting that activity of NINs was more closely related to the monkeys' trial-by-trial M/NM decisions. Together, our results suggest that PFC nonlinear integrative encoding is a key mechanism for the formation of M/NM decisions.

## PFC NINs are crucial for solving the M/NM computation in trained multi-module RNNs

RNN models trained on complex behavioral tasks have shown promise for understanding neural computations (*Engel et al., 2015*; *Masse et al., 2019*; *Song et al., 2016*)—particularly for behavioral tasks that require integrating or comparing events across time. We therefore trained RNNs to perform the DMC task in order to further examine circuit mechanisms underlying sequential decisions. Recent studies from a number of groups, including our own, have employed RNN models with a single pool of recurrent units in the hidden layer. However, this poses a challenge for relating modeling work to neuroscience questions involving multiple interconnected brain areas, as in our current study. Inspired by several recent studies (*Kleinman et al., 2019*; *Song et al., 2016*), we implemented a multi-module RNN framework, in which neurobiological principles constrain the connections between an RNN's hidden units to generate recurrently connected modules. Because our neurophysiological results suggest that MIP did not play a direct role in M/NM decisions in the DMC task, we designed and implemented RNNs composed of two hierarchically organized modules, with the module closer to sensory input intended to correspond to LIP, and the module closer to the behavioral output corresponding to PFC. The modularity and hierarchy were imposed through a set of initial constraints on networks' recurrent weight matrices, as well as the input and output weight matrices that project in sensory information and read out the behavioral decision (*Figure 8A, B*, see Materials and methods). In designing our multi-module modeling approach, we also tested several other methods of defining RNN modules, for example, via additional constraints on the identity/number of projections across modules. However, we chose the constraints used here because they consistently yielded networks whose structurally defined modules also manifested different patterns of activity and task variable encoding, suggesting effectively modular solutions to the task (see Discussion and Materials and methods).

Both modules were assigned 50% of units in the network, with matched proportions of excitatory/inhibitory neurons (80% excitatory to 20% inhibitory). The LIP module receives the motion direction input, while the PFC module projects to three response units—a unit corresponding to fixation, which the network must maintain until the test period, and separate match/nonmatch units, which simulate holding and releasing a touch-bar, respectively. Aside from these biologically derived architectural features, the modules' functional roles were not explicitly constrained (e.g., the LIP module and PFC module were not forced to encode category information or decision information, respectively). Because we found similar results using networks with hidden layers varying across a range of sizes (n = [100,200,300,400]; *Figure 8—figure supplement 1*), all results discussed henceforth used networks with n = 100 hidden units.

We independently trained 50 such networks with randomly initialized weights and identical hyperparameters to perform the DMC task using methods previously described (*Masse et al., 2019*). The DMC task that the models were trained to perform was tailored to match that used in experiments: the task sequence and motion directions were the same as in the monkey experiments, and the networks were required to indicate whether the sample and test stimuli belonged to the same category. Network parameters (recurrent weights/biases and output weights) were optimized to minimize a loss function with three parts: (1) one related to performance of the DMC task (cross-entropy of the

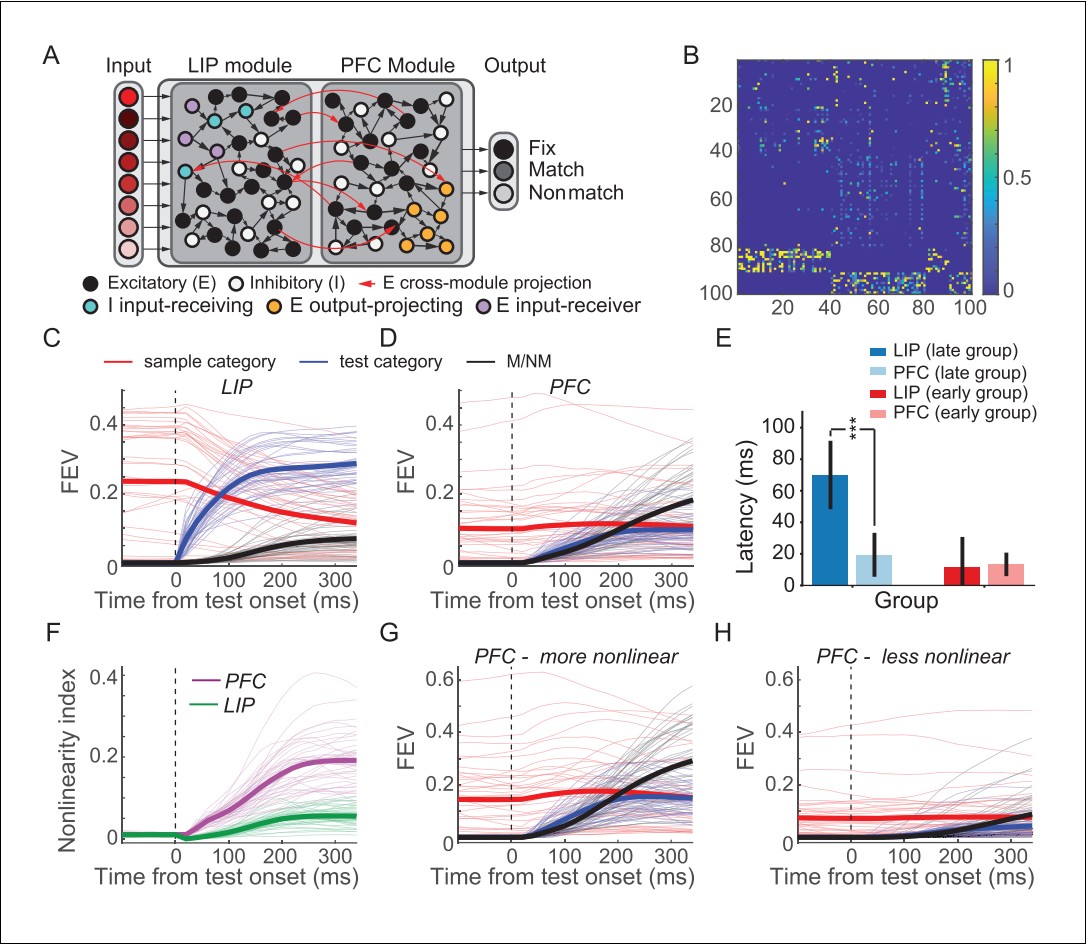

**Figure 8.** Two-module recurrent neural networks (RNNs) showed similar patterns of activity and dynamics as in neural data. (**A**) Model schematic of the two-module 'frontoparietal' RNNs. Each RNN consists of 24 motion direction turned input units, 100 hidden units, and 3 response units. The hidden layer of each RNN consists of two modules simulating lateral intraparietal (LIP) and prefrontal cortex (PFC), respectively, with half of the units designated to each module (and E/I proportion maintained). Both the excitatory and inhibitory units in each module are recurrently connected within each module. The cross-module connections are more sparse than recurrent connection within each module. Only excitatory units project to the units in the other module. (**B**) Example recurrent connectivity matrix of an example two-module RNN. Inhibition is strictly local to each module, as is emphasized by the block-diagonal structure in the bottom fifth of rows. Excitatory projections between modules are sparse, while excitatory projections within modules are denser. Each row/column represents one unit. The 1–40th and 41–80th represent PFC and LIP excitatory units, respectively; while the 81–90th and 91–100th represent LIP and PFC inhibitory units, respectively. (**C**) The averaged sample category selectivity, test category selectivity, and matching/nonmatching (M/NM) selectivity of units in the LIP modules of the 41 successfully trained RNNs were quantified using fraction of explained variance (FEV). Each thin line denotes the result from one RNN. The thick lines denote the average of all the RNNs. (**D**) The sample category selectivity, test category selectivity, and M/NM selectivity of PFC modules. (**E**) The comparison of the latencies of M/NM selectivity between LIP and PFC modules. All RNNs were separated into the late and early group based on the latency of the M/NM selectivity in LIP module. The error bar denotes STD. (**F**) The averaged nonlinearity index of units in LIP (green) and PFC (pink) modules. The thick lines denote the average across all the RNNs. (**G, H**) The task variable encodings (sample, test, and M/NM) of the more-nonlinear (**G**) and less-nonlinear (**H**) groups of units in the PFC module are shown separately.

The online version of this article includes the following figure supplement(s) for figure 8:

**Figure supplement 1.** We observe similar patterns of activity and task variable encoding across recurrent neural networks (RNNs) with different sizes (50, 100, 150, and 200 units in each module, 100, 200, 300, and 400 units for the whole networks, respectively).

network's generated outputs with respect to the correct outputs), (2) a metabolic cost on firing rates (*Harris et al., 2012*), and (3) a metabolic cost on connectivity (*Wildenberg et al., 2020*) (see details in Materials and methods). After training, 41 of 50 networks converged to perform the DMC task with high accuracy (99.9% ± 0.0009%), which were therefore included in the following analysis. Both LIP and PFC modules' units encoded all three key task variables during the test period in a similar manner to the neurophysiological data (*Figure 8C, D*). LIP module units showed significantly greater encoding of both sample and test categories (sample: p=0.0069, tstats = 2.9; test: p=7.6 × 10$^{-24}$, tstats = 21.8; df = 40, paired t-test), while PFC module units showed significantly greater M/NM encoding (p=1.2 × 10$^{-7}$, tstats = 6.4, df = 40, paired t-test). Individual units in both LIP and PFC modules encoded both sample and test category (*Figure 8—figure supplement 1*). Importantly, M/NM selectivity in the PFC module emerged with similar or shorter latency than in the LIP module. In particular, in networks where the M/NM signal emerged late (≥100 ms after test onset; n = 17 networks) in the LIP module, M/NM selectivity in the PFC module appeared with significantly shorter latency (*Figure 8E*; p=5.9 × 10$^{-9}$ tstats = 7.9, df = 16). In the other 24 networks, the latencies of M/NM selectivity in the PFC vs. LIP module did not differ significantly (*Figure 8E*; p=0.69, tstats = –0.39, df = 23), despite a minimum sensory delay of 20 ms in the PFC vs. the LIP module. As in the real LIP and PFC data, sample and test category encoding in the RNNs during the test period was more strongly correlated in the PFC than LIP module ($r_{PFC}$ = 0.34± 0.24, $r_{LIP}$ = −0.15± 0.20, p=2.3e-15, tstats = 12.5, df = 40, paired t-test), and more nonlinearly integrated in the PFC than the LIP module (*Figure 8F*, p=1.1 × 10$^{-16}$, tstats = 13.7, df = 40). Furthermore, the more-nonlinear units in PFC module showed greater encoding of all the key task variables compared to the less-nonlinear units (*Figure 8G, H*, sample: p=1.2 × 10$^{-8}$, tstats = 7.0; test: p=8.9 × 10$^{-18}$, tstats = 14.8; M/NM: p=1.2 × 10$^{-14}$, tstats = 11.8, df = 40, paired t-test). These results suggest that neural activity and information encoding in the two-module RNNs closely resembled the neurophysiological data and lend support to the idea that nonlinear integration of task-related variables in neural networks close to the output of the decision process (i.e., both PFC and the higher-order RNN module) is critical for mediating M/NM computation in the DMC task.

We next aimed to explore the circuit mechanisms underlying M/NM computation in the RNNs. Our neurophysiological data suggests that PFC is a likely site of M/NM computation, and that M/NM selectivity in LIP might be inherited from top-down input from higher areas such as PFC. To test this idea, we performed a projection-specific inactivation study in the RNNs. Specifically, we performed precise, graded ablation of top-down projections from the PFC module to the LIP module during the test period (see Materials and methods), smoothly titrating the amount of silencing applied to each projection from complete (100% reduction in efficacy) to minimal (0% reduction in efficacy). At each time point during the test period, and for each inactivation level, we measured the impact on M/NM encoding in the LIP module. If the M/NM signal that arises in the LIP module is inherited from the PFC module, then increasing abolishment of feedback from the higher to the lower module during the test period should result in an increasingly strong effect on M/NM encoding in the LIP module. In 17/41 networks, inactivating feedback during the test period had a significant effect on M/NM encoding in the LIP module (*Figure 9A*, example RNN). To quantify the correspondence between the size of the perturbation applied and the size of the effect at each time point, we computed the correlation between the inactivation extent (a vector of values between 0 and 1) and mean M/NM encoding in the module (a vector of the same length). This analysis revealed that the amount of feedback inactivation corresponded significantly with the size of the effect on M/NM encoding in the LIP module of 17/41 networks (see Materials and methods). These results suggest that M/NM encoding in the LIP module at least partially reflects top-down input from the PFC module.

To further test the causal importance of nonlinear integrative units in the PFC module during M/NM decisions, we performed an additional inactivation experiment in silico. We selectively silenced different groups of units in the PFC module during the test period and then tested the impact of that inactivation on behavioral performance of the network. For each RNN, we separated task-related units in the PFC module into two groups based on their nonlinearity index and then silenced the more-nonlinear group (top 50% ranked by nonlinearity index) and less-nonlinear group (bottom 50%) separately. This procedure allows a direct test of the causal involvement of different PFC units based on their degree of nonlinear sample-test integration. As with the projection-ablation experiments above, we smoothly titrated the amount of silencing applied to each unit from complete

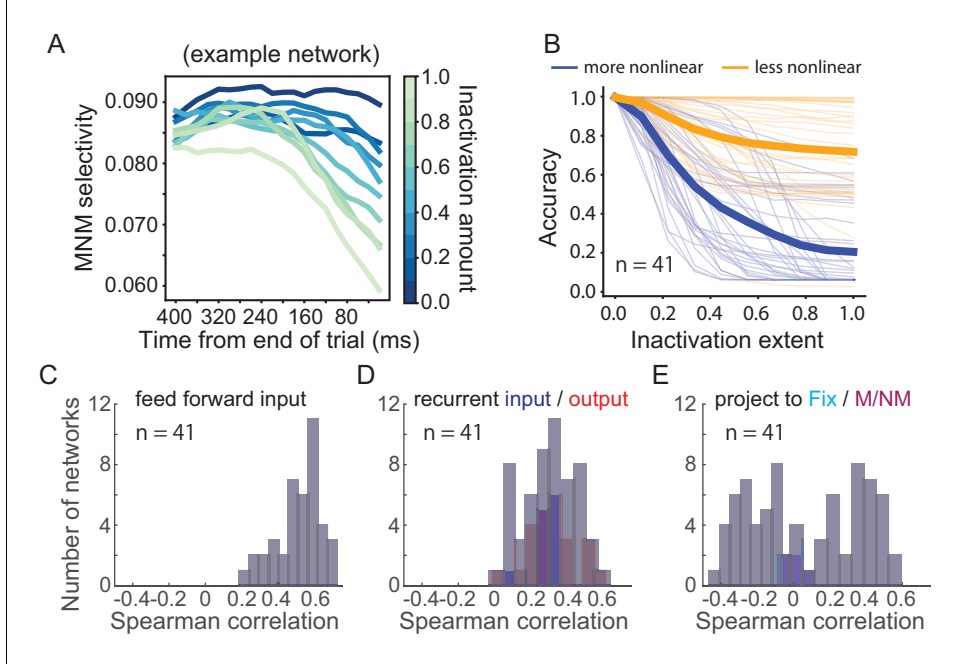

**Figure 9.** Circuit mechanisms underlying nonlinear integration of task variables to form matching/ nonmatching (M/NM) decisions. (**A**) The M/NM selectivity of units in an example lateral intraparietal (LIP) module is shown as a function of time after increasingly inactivating the feedback projection from prefrontal cortex (PFC) module during the test period. Different colors denote different inactivation levels. (**B**) The behavioral performance of the recurrent neural networks (RNNs) after gradually inactivating the more-nonlinear and less-nonlinear groups of units in the PFC module. Each thin line denotes the result from one RNN. The thick lines denote the averaged performance for all 41 RNNs. (**C**) The Spearman rank correlations between the feedforward input weights from the LIP modules and the nonlinear integrative index values of PFC module units. (**D**) The Spearman rank correlations between the recurrent connection weights of units within the PFC module and their nonlinear integrative index values. (**E**) The Spearman rank correlations between the output weights to different types of response units and the nonlinear integrative indexes of units in the PFC module. The M and NM units were responsible for reporting the match and nonmatch decisions, respectively.

The online version of this article includes the following figure supplement(s) for figure 9:

**Figure supplement 1.** Behavioral performance of recurrent neural networks (RNNs) after gradually inactivating the more-nonlinear and less-nonlinear groups of units in the prefrontal cortex (PFC) module.

(100% reduction in activity) to minimal (0% reduction in activity). At every level of inactivation, the more-nonlinear group resulted in a greater behavioral effect than the other group, even though they contained the same number of neurons and were inactivated to the same extent (*Figure 9B*). We obtained very similar results after leaving out the units directly projecting to the response units (*Figure 9—figure supplement 1*), This implicates a specific ablation of the M/NM computation rather than a gross disruption of the network's wiring to decision readouts in the behavioral deficit that results from inactivating the more-nonlinear group of PFC units. Additionally, this difference was not due to differences in the activity level of the two groups of units as the mean activity was similar between the two groups (p=0.98, tstats = 0.02, df = 40, paired t-test) during the test period. These results suggest that nonlinear integrative units in PFC module of our RNNs play a key role in the M/ NM decisions for solving the DMC task.

Lastly, we explored potential circuit mechanisms underlying this critical role of nonlinear integrative encoding in the PFC module during M/NM decisions. Because they allow complete knowledge of the connectivity between hidden units, RNNs are also a model system uniquely well-suited to exploring circuit mechanisms that underlie network behavior. We examined the input and output weights of the PFC units and calculated the correlation between the connection weights and the degree of nonlinear integrative encoding of the PFC units for each RNN. We found that PFC nonlinear integrative units were more likely to receive greater feedforward input from the LIP module,

indicated by the positive correlations between the nonlinearity index and the feedforward input weights from the LIP module (*Figure 9C*, r = 0.52 ± 0.13). Meanwhile, the nonlinear integrative units were more recurrently connected within the PFC module than the other PFC units, indicated by the positive correlations between PFC neurons' nonlinearity index and their recurrent input/output weights (*Figure 9D*, input: r = 0.27 ± 0.14; output: r = 0.34 ± 0.14). Furthermore, the nonlinear integrative units in the PFC module were more likely than other PFC module units to project to the match and nonmatch response units, but not the fixation response unit. This is shown by the positive correlations between the nonlinearity index and the output weights to the M/NM response units, and negative correlations between the nonlinearity index and the output weights to the fixation unit (*Figure 9E*, M/NM units: r = 0.30 ± 0.17; fix units: r = −0.20 ± 0.13). These results suggest that PFC nonlinear integrative units may derive their nonlinear encoding as a result of their hub-like strong interconnection with both input and output units as well as recurrent connections with other PFC module units.

Together, the two-module RNN simulations and in silico inactivation experiments reinforce the plausibility of our central neurophysiological findings of the importance of PFC nonlinear integrative encoding of task-relevant information for mediating sequential decisions during the DMC task.

## Discussion

In this study, we directly compared neural activity in PFC, LIP, and MIP in monkeys performing a delayed match-to-category task and focused on understanding how sequential decisions are carried out across the cortex. In particular, we sought to understand where, when, and how the remembered sample stimulus and currently visible test stimulus are integrated to reach a M/NM decision. We found that PFC functions as a candidate source of M/NM decision signals. By contrast, LIP's role is more aligned with stimulus evaluation and short-term memory, and MIP primarily reflects premotor/motor functions. We also highlight a particular form of encoding in PFC—nonlinear mixed encoding of sample and test information—during the decision phase of the task as being especially important for mediating the monkeys' M/NM decisions.

These interpretations arise primarily from comparisons of the magnitude, latency, and format with which task variables are encoded in PFC, LIP, and MIP. First, we found that test-period activity in LIP showed the strongest categorical encoding of both the remembered sample and the currently visible test stimulus, consistent with it playing a primary role representing the category of both visible and remembered stimuli during the DMC task. Second, LIP appears less directly involved than PFC in transforming categorical encodings into M/NM decisions. This is supported by the longer-latency M/NM selectivity in LIP compared with PFC, suggesting that M/NM encoding in LIP may reflect input from higher cortical areas, such as PFC. Test-period activity in MIP, on the other hand, was found to reflect only the remembered sample category but not the currently visible test category. Instead, MIP activity during the test period was dominated by motor-related encoding arising with a longer latency compared to M/NM selectivity in PFC, consistent with previous reports (*Cui and Andersen, 2007*; *Swaminathan et al., 2013*). These results indicate that MIP is unlikely to be directly involved in the comparison of sample and test categories, but instead may receive M/NM signals from another decision-related area, such as PFC, during decision execution.

Our analyses also reveal three lines of evidence that PFC leads the M/NM decision process during sequential decision tasks like DMC. First, M/NM selectivity of both spiking and LFP signals arose with a shorter latency in PFC than in both PPC areas, consistent with a flow of M/NM encoding from PFC to PPC. Second, PFC neurons showed a relatively balanced preference for both 'match' and 'nonmatch,' while LIP and MIP were biased toward preferring 'match' conditions, which were accompanied by hand movements. Balanced M/NM representation in PFC suggests it is more likely to reflect the abstract M/NM decision rather than preparatory motor activity. Furthermore, those PFC neurons responding more strongly to nonmatching test stimuli may be involved in PFC's established role in response inhibition (*Aron et al., 2014*; *Krämer et al., 2013*; *Schall and Godlove, 2012*) (i.e., withholding a motor response on nonmatch trials). Third, the sample and test categories are combined more nonlinearly in PFC than in LIP and MIP, and PFC neurons showing nonlinear encoding were more strongly correlated with the monkeys' decisions than other PFC neurons. Previous studies using a delayed match to sample task with visual motion stimuli observed comparison-related activity in both PFC and medial temporal cortex (MT), but found that such activity was decision-

correlated only in PFC (*Lui and Pasternak, 2011*; *Zaksas and Pasternak, 2006*). Together, these results are suggestive of an abstract M/NM decision process in PFC that intervenes between stimulus evaluation (motion categorization) in areas like LIP (*Zhou and Freedman, 2019*) and motor planning in areas like MIP.

Previous studies using a delayed match to sample task, rather than the categorization task used in our study, reported that some MT neurons' activity was suppressed on match trials, and that match suppression in MT emerges with a shorter latency than M/NM selectivity in PFC. This suggests that a distributed network, including both PFC and early sensory cortex, might be involved in sequential decisions (*Hussar and Pasternak, 2012*). Although we observe M/NM selectivity across frontoparietal cortical areas, the pattern of results we observed suggests that PFC is a likely site of M/NM computation in the DMC task. LIP is considered to be closer to sensory input and upstream motion processing areas (e.g., MT) compared to PFC, but our results suggest that PFC is more closely involved in M/NM computation than LIP. Different conclusions regarding the roles of these areas between different studies could be due to the unique cognitive demands of the DMC task compared to the delayed match to sample tasks used in previous studies, which could lead to differences in the network of areas recruited to solve each task. Previous work in MT during the DMC task did not find abstract encoding of visual motion categories (*Freedman and Assad, 2006*). This makes it unlikely that MT would show categorical match enhancement or suppression during the DMC task.

Cortical neurons have been shown to encode mixed representations of multiple task variables during cognitively demanding tasks (*Johnston et al., 2020*; *Parthasarathy et al., 2017*; *Rigotti et al., 2013*; *Zhang et al., 2017*), and nonlinear mixed selectivity (NMS) in PFC has been particularly emphasized as an important mechanism for cognitive computations. Specifically, NMS can potentially facilitate a linear readout of task variables, and the strength of NMS is correlated with the subjects' behavior (*Fusi et al., 2016*; *Ramirez-Cardenas and Viswanathan, 2016*; *Rigotti et al., 2013*). Our observation of nonlinear integrative encoding in PFC may be related to recent experimental and theoretical work on NMS in PFC (*Parthasarathy et al., 2017*; *Rigotti et al., 2013*; *Zhang et al., 2017*). Our results suggest that the way in which task variables are integrated (i.e., linear vs. nonlinear) differs between cortical areas, and such differences potentially give insights into underlying functions of each area. We found that test period encoding of the remembered and currently visible stimuli was integrated, or mixed, in both LIP and PFC, but in different ways. In LIP, the linearly integrated encoding of sample and test information could faithfully encode stimulus information and facilitate downstream readout of both variables, which is in accordance with LIP's role in evaluating sensory stimuli (*Zhou and Freedman, 2019*). In contrast to LIP, sample and test category selectivity was more nonlinearly integrated in PFC, and such nonlinear integrative encoding was correlated with the monkeys' M/NM decisions. These results build on and extend previous findings, and suggest that nonlinear integrative encoding in PFC is a key mechanism for manipulating the encoding of sensory stimuli and items in working memory to form decision-related (M/NM) representations. Our results are consistent with the idea that mixed integrative encoding in PFC and LIP may relate to the core functions of each area during the DMC task: in areas closer to sensory input, such as LIP, linear integrative encoding may support independent encoding of visible and remembered stimulus features. In areas more associated with cognitive or task-related functions, such as PFC, NMS may facilitate the integration of task-relevant variables in order to satisfy the task demands.

Despite neurophysiological evidence that PFC neurons nonlinearly mix/integrate information, a mechanistic, circuit-level understanding of how they do so has remained elusive. Here, we extended a burgeoning class of model system—artificial RNNs trained to perform cognitive tasks—to explore the mechanisms for and importance of such nonlinear integration during decision making (*Masse et al., 2019*; *Song et al., 2016*). Several recent efforts have successfully trained such biologically inspired RNNs with multiple modules (analogous to distinct cortical areas) (*Kleinman et al., 2019*; *Michaels et al., 2020*; *Pinto et al., 2019*). Inspired by these, we trained a population of hierarchical, modular RNNs, the lower corresponding with LIP and the higher with PFC, to perform the DMC task. The units of our two-module RNNs exhibited highly similar patterns of activity as in neuronal physiology data, including both the key features of information encoding within either LIP or PFC and the functional differences between LIP and PFC. Future studies with RNNs that include more realistic motor output modules will likely be useful for understanding how decisions are transformed into specific actions.

These RNN models' primary benefits are twofold: first, they offer full knowledge of network connectivity; and second, they facilitate causal perturbations analogous to those used in experimental preparations. Using the RNNs' synaptic connectivity matrices, we first identified a relationship between neuronal connectivity and nonlinear integration. Relative to PFC module units that more linearly integrated sample/test category information, nonlinear integrative units in the PFC module were more strongly connected to inputs, outputs, and with one another. This suggests that PFC NINs might serve as 'hubs' for information integration and transformation to solve cognitive tasks. We also extended causal approaches used in neuroscience experiments to provide important insights into the mechanisms underlying M/NM computation and the role of nonlinear integration in the trained RNNs. In vivo, perturbations of neuronal activity/connectivity are usually targeted based on features like anatomy or genetic identity. Here, we adopted an alternative targeting approach, selectively and gradually silencing the activity of artificial units based on the extent to which they show a particular pattern of information encoding (in this case, nonlinear integration of sequentially presented stimuli), and ablating connections based on the identities of the pre- and postsynaptic modules. Using this type of functional targeting, we first examined the functional interaction between the LIP and PFC modules, finding that M/NM selectivity in the LIP module was significantly modulated by top-down input from the PFC module. Second, we validated the necessity of nonlinear integrative encoding for mediating the M/NM decisions during the DMC task. These are important complements to our experimental results, which demonstrate correlations between the activity of different neurons in PFC/LIP with monkeys' trial-by-trial decisions, and make predictions which can guide future experiments and data analysis.

Critically, however, we note that not all of the approaches we tried for enforcing modularity by constraining connectivity resulted in networks that exhibited functionally modular solutions to the DMC task. We found that when the criteria that define which neurons can send or receive out-of-module connections are too restrictive, for example, networks are not allowed sufficient flexibility in how they communicate information between modules. In this regime, networks can be pushed to 'solve' tasks within one module rather than effectively modularizing the computation across all the modules, and thus appear modular in structure but not in function. Careful verification that modular connectivity also results in modular computation, through analysis of RNN activity and encoding, will therefore be essential when using multi-module RNN models in studying neural processes distributed across multiple brain regions. In our experiments, we found parsimony to be a useful principle in obtaining functionally modular networks, and had the most success using a simple set of constraints—sparser connections between modules than within modules, with inhibitory connections restricted to within-module targets. Although we also explored additional restrictions on how modules communicate, including the prohibition of direct connections between units receiving external sensory inputs and units projecting to the network's behavioral output, these additional restrictions did not help to yield networks matching the key features of the neural data in the current study.

It will be important to extend this work to examine the roles played by a wider network of cortical and subcortical areas in sequential decisions. This includes premotor cortex, which shows decision-related activity during a shape (cat vs. dog) DMC task (*Cromer et al., 2011*) and abstract decision tasks (*Wallis and Miller, 2003*), as well as subcortical structures such as basal ganglia and thalamus. Future studies need to conduct large-scale simultaneous recordings from multiple brain areas to characterize the real-time functional interactions among these brain regions, as well as extend investigations of multi-module RNNs in parallel with experimental work.

## Materials and methods

### Datasets

This study includes five datasets from two independent experiments. Most of the data are from a DMC experiment that includes a PFC dataset, an LIP dataset, and an MIP dataset. Analyses from these datasets have been published previously (*Swaminathan and Freedman, 2012*; *Swaminathan et al., 2013*), though unrelated to the present study. The data in *Figure 3—figure supplement 1* originated from a DMC learning experiment (*Masse et al., 2017*). This study was performed in strict accordance with the recommendations in the Guide for the Care and Use of Laboratory Animals of the National Institutes of Health. All of the animals were handled according to

approved Institutional Animal Care and Use Committee (IACUC) protocol of The University of Chicago.

## Behavioral task and stimulus display

The DMC task has been described previously (*Freedman and Assad, 2006*; *Swaminathan and Freedman, 2012*) and is briefly summarized below. In this task, monkeys were trained to release a lever when the categories of sequentially presented sample and test stimuli matched, or hold the lever when the sample and test categories did not match. Stimuli consisted of six motion directions (15°, 75°, 135°, 195°, 255°, 315°) grouped into two categories separated by a learned category boundary oriented at 45° (*Figure 1B*). Trials were initiated by the monkey holding the lever and keeping central fixation. Monkeys needed to maintain fixation within a 2° radius of a fixation point through the trial. 500 ms after gaze fixation was maintained, a sample stimulus was presented for 650 ms, followed by a 1000 ms delay and a 650 ms test stimulus. If the categories of the sample and test stimuli matched, monkeys needed to release a manual touch-bar within the test period to receive a juice reward. Otherwise, monkeys needed to hold the touch-bar during the test period and a second delay (150 ms) period, and wait for the second test stimulus, which was always a match, and then release the touch-bar, so that monkeys concluded all trials with the same motor response (lever release). The motion stimuli were high contrast, 9° diameter, random-dot movies composed of 190 dots per frame that moved at 12°/s with 100% coherence. Task stimuli were displayed on a 21-inch color CRT monitor (1280 * 1024 resolution, 75 Hz refresh rate, 57 cm viewing distance). Identical stimuli, timing, and rewards were used for both monkeys in all PFC, LIP, and MIP recordings. Monkeys' eye positions were monitored by an EyeLink 1000 optical eye tracker (SR Research) at a sampling rate of 1 kHz and stored for offline analysis. Stimulus presentation, task events, rewards, and behavioral data acquisition were accomplished using an Intel-based PC equipped with MonkeyLogic software running in MATLAB (*Asaad et al., 2013*) (http://www.monkeylogic.net).

In the DMC learning experiment, two other monkeys were trained to perform a slightly altered version of the standard DMC task. Identical setups, stimuli, timing, and rewards were used; however, only 24 stimulus conditions (sample-test-direction combinations) were used. Neuronal activity was recorded while monkeys learned this DMC task, which is after the monkeys had learned a delayed match to sample (direction) task.

## Electrophysiological recording

Two male monkeys (*Macaca mulatta*, 8–10 kg) were implanted with a head post and recording chambers positioned over PPC and PFC. Stereotaxic coordinates for chamber placement were determined from magnetic resonance imaging (MRI) scans obtained before chamber implantation. PFC chambers were centered on the principal sulcus and anterior to the arcuate sulcus, ~27.0 mm anterior to the intra-aural line. Areas LIP and MIP were accessed from the same PPC chamber, which was positioned over the intraparietal sulcus (IPS) centered ~3.0 mm posterior to the intra-aural line. All experimental and surgical procedures were in accordance with the University of Chicago Animal Care and Use Committee and National Institutes of Health guidelines. Monkeys were housed in individual cages under a 12 hr light/dark cycle. Behavioral training and experimental recordings were conducted during the light portion of the cycle.

LIP and PFC recording sessions were interleaved in each monkey to reduce the influence of timing on the neuronal responses and monkeys' behavior. In monkey A, 35 PFC recordings sessions were followed by 29 LIP sessions and an additional 15 PFC sessions. In monkey B, most LIP recordings (n = 22 sessions) were conducted first, followed by PFC recordings (n = 36 sessions) and simultaneous LIP-PFC recording sessions (n = 4 sessions). The MIP recordings were conducted in separate sessions after completing PFC and LIP recording.

All recording equipment and procedures were the same as in the previous studies (*Swaminathan and Freedman, 2012*; *Swaminathan et al., 2013*). LIP and MIP recordings were conducted using single 75 µm tungsten microelectrodes (FHC), a dura piercing guide tube, and a Kopf (David Kopf Instruments) hydraulic micro-drive system. In general, LIP cells were found at more lateral locations and MIP cells were found at more medial locations within the same recording chamber. LIP was 2–7 mm below the surface and MIP was 1–5 mm below the surface in both monkeys. PFC recordings were made using 250 µm dura-piercing tungsten microelectrodes (FHC) and a

custom manual micro-drive system that allowed simultaneous recordings from up to 16 electrodes. Neurophysiological signals were amplified, digitized, and stored for offline spike sorting (Plexon) to verify the quality and stability of neuronal isolations. The offline spike sorting used the same standard as in the previous studies, which ensured that each single neuron was well isolated.

In the DMC learning experiment, two additional monkeys (*M. mulatta*, 9–12 kg) were implanted with a head post and two 32-channel semi-chronic recording systems (Gray Matter Research) on PPC and PFC. MRI scans were used to guide chamber placement. For PPC recordings, chambers were placed over the IPS, ~2.0 mm posterior to the intra-aural line and ~14.0 mm lateral from the midline for monkey Q, and ~2.0 mm anterior to the intra-aural line and ~13.0 mm lateral from the midline for monkey W. We advanced all PPC electrodes until their estimated positions were below the IPS, guided by its known anatomical depth. Additional evidence for electrode depth on many recording channels was the marked reduction in spiking activity as electrodes entered the sulcus. For PFC recordings, chambers were placed over the principal sulcus, ~29.0 mm anterior to the intra-aural line and ~20.0 mm lateral from the midline for monkey Q, and ~33.0 mm anterior to the intra-aural line and ~22.0 mm lateral from the midline for monkey W. Each micro-drive system contained 32, 125 µm tungsten microelectrodes (Alpha-Omega). Before each session, we lowered electrodes between 0 and 1 mm to optimally record the spiking activity of well-isolated neurons. Neuronal activity in PFC and PPC was recorded simultaneously for every session. The PPC recording might include both LIP and MIP. We used the same standard for offline isolation of single neuron as in the regular DMC experiment. All experimental and surgical procedures were standard and in accordance with the University of Chicago Animal Care and Use Committee and National Institutes of Health guidelines.

## Receptive field mapping and stimulus placement

All PFC and LIP neurons as well as most MIP neurons were tested with a memory-guided saccade (MGS) task before DMC task. LIP neurons were identified by spatially selective visual responses and/or persistent activity during the MGS task. MIP neurons were identified by responses during the animals' spontaneous hand movements, such as lever releases, scratching, or arm movements observed before the DMC task commenced, and the absence of modulation during the MGS task. LIP and MIP neurons were also differentiated based on anatomical criteria, such as the location of each electrode track relative to that expected from the MRI scans, the pattern of gray–white matter transitions encountered on each electrode penetration, and the relative depths of each neuron.

Motion stimuli for the DMC task were always targeted to LIP receptive fields (RFs). The typical eccentricity of stimulus placement for LIP recordings was ~6.0–10.0°. During MIP recordings, the motion stimulus was always placed at 7° from the fixation along the horizontal axis contralateral to the recording hemisphere. For most PFC recordings (n = 55 of 86 sessions), sample and test stimuli were presented in blocks of 30 trials at three nonoverlapping locations in the contralateral visual field centered 7.0° from fixation, which covered much of the contralateral visual field on the monitor. For the remaining PFC recording sessions (31 of 86), stimuli were shown at a single fixed location, 7.0° from fixation along the horizontal axis in the contralateral visual field. All recorded trials for PFC neurons were used for subsequent analyses. Similar results were observed using only the one-location or three-location PFC datasets, or using PFC data for which stimuli were presented at the best of the three locations. None of the recorded neurons were pre-screened for direction, category, or M/NM selectivity before recording.

In the DMC learning experiment, we recorded all neurons with well-isolated action potentials, as we could not place stimuli within the RFs of all recorded neurons. To increase the chances that stimuli were in or near neuronal RFs, we ran the experiment in alternating 10-trial blocks in which stimulus position was varied between two nonoverlapping positions (7.0° eccentricity; ±45° relative to horizontal meridian) in the visual field that was contralateral to the hemisphere targeted for neuronal recordings. Analysis of neuronal data revealed qualitatively similar results (in both cortical areas) for each of the two stimulus locations considered separately or when trials from the two locations were combined. Thus, we combined trials for both stimulus locations in current study.

## Data analysis

### Pre-analysis neuron screening

We used multi-electrodes to record PFC neurons and did not pre-screen neurons prior to recording. For LIP and MIP recording, we used single-electrode recording and applied some standard criteria to screen the neurons (visual responsiveness for LIP and movement responsiveness for MIP). Thus, the total number of neurons was much larger in PFC than in LIP and MIP (PFC: 447; LIP: 75; MIP: 94). However, many PFC neurons exhibited very low firing rates and were not task-modulated during the task interval, and therefore might not contribute to the task variable representations. In contrast, most of the recorded LIP and MIP neurons showed relatively high firing rates and were task modulated. To reduce any potential confounds that might be caused by differences between PFC and PPC (LIP and MIP) datasets, we used the following criteria to further pre-screen all neurons for data analysis: (1) the maximum of the mean conditional averaged firing rate during the task interval (from fixation onset to 350 ms after test onset) should be no less than 5 spikes/s; and (2) the activity should exhibit at least one kind of task-related modulation (such as sample category selectivity, test category selectivity, and M/NM selectivity, one-way ANOVA test, $p<0.01$) during one of the four task intervals (sample period, earlier delay period, later delay period, and test periods). After screening, 145 PFC neurons, 53 LIP neurons, and 66 MIP neurons were included for further analysis. We also tested different thresholds (1 spike/s or 4 spike/s) to screen the neurons, which produced similar results.

In order to select neurons that showed significant M/NM selectivity during the test period, we applied a one-way ANOVA test ($p<0.01$) to the mean activity within a 200 ms time window, sliding by 5 ms, during the test period (50–350 ms after test stimulus onset). To compare M/NM selectivity time courses across different cortical areas (*Figure 3*), we only selected neurons that showed significant M/NM selectivity during the early test period (50–300 ms after test onset). The results were qualitatively similar when we used different time windows (50–250 or 50–350 ms after test onset) to select neurons.

In the DMC learning experiment, we used the same criteria with one difference to screen neurons for the data analysis. Since most neurons exhibited a very low firing rate, we selected the neurons that had a maximum firing rate greater than 4 spike/s during the task interval to include more neurons. We also used different thresholds (1 spike/s or 5 spike/s) to screen neurons and obtained similar results.

### Behavioral performance quantification

For all recording sessions that contained trials for all 36 stimulus conditions, we calculated the monkey's accuracy for each condition within a single session and then averaged across all sessions. To compare performance across PFC, LIP, and MIP datasets, we first calculated the overall average accuracy for each session and then applied a one-way ANOVA test to test whether there were any differences between different datasets.

We separated both match trials and nonmatch trials into easier and more-difficult subgroups based on sample and test motion directions as well as monkey's averaged performance across all recording sessions (including all PFC, LIP, and MIP data) for all the 36 stimulus conditions separately for each monkey. There were 10 stimulus conditions in which the motion direction of either sample or test or both stimuli were center direction (135˚ or 315˚) for both match and nonmatch trials. We defined 9 of the 10 stimulus conditions in which monkeys showed higher average accuracy for both match and nonmatch trials as easier subgroup and the other nine stimulus conditions as more-difficult subgroup. Thus, there were roughly an equal number of trials between the easier and more-difficult subgroups for each sample and test category. To correlate M/NM selectivity with monkeys' RT, we separated the match trials into faster and slower RT trials (below or above median RT) for all conditions in each session; the faster and slower RT trials from two category conditions were pooled together.

### Spike density function and normalized activity

For all the figures showing the activity of example neurons and population neurons, we used a 20 ms Gaussian window to smooth the peristimulus time histogram (PSTH). In *Figures 2* and *4*, the activity of each neuron was normalized by its maximum firing rate.

## Equating (decimating) firing rates

We equated the firing rate between match-preferring and nonmatch-preferring neurons when their M/NM selectivity was compared. For each brain area, we first computed a ratio (R) of the averaged firing rate of the nonmatch-preferring neurons over the averaged firing rate of match-preferring neuron during the test period. Since the averaged firing rate of match-preferring neurons is higher than that of nonmatch-preferring neurons in all three cortical areas, for each match-preferring neuron, we then randomly removed from respective spike trains a number of action potentials that corresponded to the rounded product of 1 R with the number of action potentials for each trial.

## ROC analysis

We applied ROC analysis to the distribution of firing rates (50 ms sliding time window with a 5 ms step) of each neuron during the test period to quantify their M/NM selectivity. The area under the ROC curve is a value between 0.0 and 1.0 indicating the performance of an ideal observer in assigning M/NM choice based on each neuron's trial-by-trial firing rates. Values of 0.0 and 1.0 correspond to strong encoding preference for nonmatch or match, respectively. Values of 0.5 indicate no M/NM selectivity.

To test whether the M/NM selectivity in PFC (*Figure 7*) correlated with monkeys' trial by trial choice, we used ROC analysis to evaluate the activity change in error-match trials relative to correct match trials. Since the number of correct trials greatly exceeded error trials and might influence the reliability of ROC values, we applied a shuffling procedure to equalize the trial totals between correct and error trials. We first randomly selected the same amount of correct trials as the error trials and calculated the ROC value. Then, we repeated this procedure 100 times and averaged the 100 ROC values. The ROC values were calculated in slightly different ways for match-preferring and nonmatch-preferring neurons: values greater than 0.5 indicate that neuronal activity on incorrect match trials was more similar to activity on correct nonmatch trials, that is, lower activity in error-match trials than in correct-nonmatch trials for match-preferring neurons, or greater activity in error-match trials than in correct-nonmatch trials for nonmatch-preferring neurons. This is consistent with a correlation between neurons' M/NM selectivity and monkeys' trial-by-trial M/NM choices. ROC values near 0.5 indicate similar activity between incorrect match and correct match trials, which indicates that neuronal activity is not correlated with the monkeys' trial-by-trial choices. ROC values lower than 0.5 indicate even greater M/NM selectivity between incorrect match and correct nonmatch trials.

## Unbiased FEV

To quantify M/NM and category selectivity, we performed one-way ANOVA on the neuron's average firing rate within a sliding window (width = 50 ms, step size = 5 ms) using either the M/NM choice or the category membership as factors. To quantify the amount of information that a neuron encoded about each factor that was independent of the absolute neuronal firing rate, we calculated the unbiased FEV in the neuron's firing rate that could be attributed to the M/NM choice or category membership (sample category or test category) with the following:

$FEV_{factor} = (SS_{factor} - (k-1)MSE)/(SS_{total} + MSE)$, where $SS$ indicates the sum of squares, MSE indicates mean square error, and k indicates number of conditions.

We also calculated the unbiased FEV of M/NM choice for the LFP signal. We directly applied the analysis on the average amplitude of the LFP signal within a 50 ms sliding window in all recording channels of each cortical area. The LFP signal was pre-filtered by a bandstop filter (Butterworth, 59~61 HZ) to remove power-line noise.

## ROC-based category tuning index (rCTI)

We used the rCTI measurement to quantify the category selectivity, which was described in detail in our previous work (*Swaminathan et al., 2013*) and defined as follows: rCTI = BCD - WCD,

WCD = (2 |ROC(75,195) - 0.5| + |ROC(135,195) - 0.5| + |ROC(75,135) - 0.5| + 2|ROC(255,15)- 0.5|+ |ROC(315,15) - 0.5| + |ROC(255,315) - 0.5|)/8,

BCD = (2|ROC(75,15) - 0.5| + |ROC(75,315) - 0.5| + |ROC(135,255) - 0.5| + |ROC(135,15) - 0.5|+ 2| ROC(195,255) - 0.5| + |ROC(195,315) - 0.5|)/8.

### Latency of M/NM selectivity

We calculated the latency of M/NM selectivity for spiking activity or LFP signals using the following criteria: (1) the activity (spike count or mean LFP amplitude) in at least two successive (following) sliding time windows (20 ms width, 20 ms step size) showed significant M/NM selectivity (one-way ANOVA test, $p < 0.05$); and (2) the M/NM preference of the activity in these time windows must be consistent with the global M/NM preference during the test period (50–350 ms after test onset). For this calculation, we only analyzed neurons that showed significant M/NM selectivity. When we compared the latency between different subgroups or brain regions, we used only the neurons for which we could calculate a latency according to these criteria. We also tested different criteria (different numbers of sliding window such as 5) to perform the analysis and obtained similar results.

### Correlation between sample category and test category representation and selection of neurons showing mixed category selectivity

To test the correlation between sample and test category selectivity during the test period, we performed two-way ANOVA on the neuron's activity and calculated the unbiased FEV of both sample category and test category as mentioned above during the early test period (0–250 ms after test stimuli onset), which mostly preceded the monkeys' decision. The analysis was applied on the mean firing rates of each neuron with a 100 ms sliding window (5 ms step). The maximal FEV value of sample category and the maximal FEV value of test category of each single neuron were chosen to calculate the rank correlation between sample and test category selectivity in each cortical area. Neurons that showed significant selectivity for both sample and test category in the selected sliding windows above (0~250 ms after test onset, $p < 0.01$) were defined to be mixed category-selective neurons.

### SVM decoding

Similar to previous studies (*Sarma et al., 2016*), we used a linear SVM classifier to decode monkeys' M/NM choice, category membership, and sample-test-category combination separately from a surrogate population of three cortical areas. In this surrogate population, activities from different neurons in one cortical area were treated as if they were recorded simultaneously although neurons were, for the most part, not recorded simultaneously. The linear classifier was trained using an SVM. In training a linear classifier, a hyperplane that best separates the trials belonging to two or several different classes was determined. In the case of decoding M/NM choice, each class corresponds to one type of choice. In contrast to previous studies (*Swaminathan et al., 2013*), we used all six motion directions together to perform the decoding analysis for category as we think the direction turning might also contribute to the category representation.

Decoding was applied to the mean firing rates of neurons within a 50 ms sliding window (5 ms step). For each neuron, we randomly selected 66% of trials to train the classifier and left the other 34% of trials for testing. We then randomly sampled, with replacement, 120 trials from the training list and 60 trials for the testing list for bootstrapping. In order to reduce the potential confound caused by uneven number of trials of different motion directions, a minimum number for trials of each motion direction was required for random sampling (10 and 5 trials for training and testing data of each direction, respectively). To compare different types of selectivity across different populations, we applied a shuffling procedure to select an equal number of neurons for all decoding analyses except in *Figure 4*. We bootstrapped all decoding analyses 100 times.

### Definition of nonlinear integrative encoding

We used two-way ANOVA to identify linearly and nonlinearly integrated sample and test category representations as a previous study (*Lindsay et al., 2017*). This method is conceptually similar to the approaches used in the early study on NMS (*Rigotti et al., 2013*). Specifically, we defined the nonlinear integrative neurons as those that exhibited significant sample and test category selectivity, as well as significant interaction between sample and test category selectivity. Meanwhile, we identified the neurons that exhibited a significant interaction between sample and test category selectivity, but nonsignificant sample and test category selectivity, as pure M/NM-selective neurons. We did not classify the pure M/NM-selective neurons as mixed selective because they exhibited neither significant sample category selectivity nor significant test category selectivity. Therefore, the interaction term cannot be directly related to the NMS in our study. However, the structure of the task used in

the current study prohibits us from following their analytical approach due to the smaller number of distinct conditions tested in our task (*Fusi et al., 2016*; *Ramirez-Cardenas and Viswanathan, 2016*; *Rigotti et al., 2013*).

## Nonlinearity index

To quantify the nonlinearity of mixed sample and test category selectivity, a nonlinearity index was calculated for each mixed category-selective neuron in PFC and LIP. The nonlinearity index was defined as the test category selectivity difference between two sample category conditions:

NLI = | FEV $_{(S1T1\ vs.\ S1T2)}$ - FEV $_{(S2T1\ vs.\ S2T2)}$|; where NLI indicates nonlinearity index, and '| |' indicates absolute value. If the test category selectivity is linear addictive to the existing sample category selectivity, or the neuron purely responds to the M/NM status of the test stimuli, then FEV$_{(S1T1\ vs.\ S1T2)}$ = FEV$_{(S2T1\ vs.\ S2T2)}$. Therefore, if there is pure linear mixed sample-test-category selectivity or pure M/NM selectivity, the value of nonlinearity index will be 0. According to the above formula, the value of the nonlinearity index would correlate with the test category selectivity as neurons showing greater test category selectivity would potentially show greater nonlinearity index. In order to diminish the potential confound caused by the test category difference between LIP and PFC, we normalized the nonlinearity index of each neuron to the averaged value of test category selectivity in PFC and LIP, respectively, when compared to the nonlinearity index between two areas.

## Recurrent neural network training

All RNN analyses involved training biologically inspired networks, as described previously (*Masse et al., 2019*). These differ from standard RNNs in two main ways: first, they contain separate excitatory and inhibitory units, per Dale's law; and second, synapses are endowed with short-term plasticity, allowing synaptic efficacies to fluctuate over short timescales in an activity-dependent manner, as in a previous study from our group. All networks were trained using Tensorflow (*Martín Abadi et al., 2016*) on a GEFORCE RTX-2080Ti GPU with the same hyperparameters: 100 hidden units, 80 of which were excitatory and 20 of which were inhibitory; a learning rate of 0.01; a batch size of 256 trials; metabolic costs on mean activity and mean connection strength, weighted consistently across all networks; and initial weights and biases for input, hidden, and output weight matrices drawn from identical distributions across all networks. All networks received input from 24 motion direction-tuned neurons, with tuning distributed according to a von Mises function with concentration factor 2 and a scaling factor of 4. All networks were wired to two output units, which corresponded with the animals' behavior of maintaining fixation, indicating match, and indicating nonmatch on each trial.

The E/I ratio (80% excitatory to 20% inhibitory) was chosen to be consistent with the range of proportions found in the mammalian neocortex (*Markram et al., 2004*), and accordingly reflects the standard among models constrained to obey Dale's principle. Further, in this kind of network, a similar amount of inhibition is generally required for stable convergence. Networks where excitation and inhibition are inappropriately balanced undergo runaway activity, and thus are difficult to train using gradient-descent methods. Network hidden neurons used the ReLU activation function, which is linear and nonsaturating for non-negative activities and clips negative activities to 0. Input projections were linear and fixed (e.g., not trainable, to prohibit networks from discarding motion direction information immediately). The output layer (fixation, match, and nonmatch units) used the softmax activation function, which scales output unit activities to generate a probability distribution over output values at each time point. Network parameters (recurrent weights/biases and output weights) were optimized to minimize a loss function with three parts: (1) a performance loss, given by the categorical cross-entropy between the desired vs. actual output activities, which pushes networks to perform the DMC task at a high level of accuracy; (2) a metabolic cost on firing rates (mean neuronal activity), which pushes networks to solve the task without using firing rates that are pathologically high (*Harris et al., 2012*); and (3) a metabolic cost on connectivity (mean synaptic weight), to reflect the costliness of maintaining synaptic connections in vivo (*Wildenberg et al., 2020*).

For each trial of the DMC task, networks were presented with a sample motion direction stimulus; following a stimulus-free delay period, the networks were presented with a second motion direction, after which they were required to determine whether the test was a categorical match to the sample. Importantly, each element of task design in the DMC task the models were trained to perform was

tailored to match those used in the monkey experiments: in particular, sample/delay/test epochs were of the same duration used in experiments, and motion stimuli were drawn from the same set of motion directions. Trials were programmatically generated by choosing sample/test stimulus identities and using these to construct inputs to the networks at each timestep (based on the motion tuning of the input layer) and desired output (fixation, match, nonmatch) at each timestep.

We trained 50 networks to perform DMC (*Masse et al., 2019*). 41/50 networks achieved consistently high performance by the end of training (>95% accuracy for the last 50 batches). In order to verify that our RNN modeling results did not critically depend on the number of neurons, we trained more example networks with different numbers of neurons (100, 200, 300, 400) using the same constraints for modularity used in the rest of the article (*Figure 8—figure supplement 1*). We found that the key results are consistent between the networks with different sizes, confirming that the results we obtained are robust to network size within a reasonable range, and thus that network size was not the key constraint for determining the characteristics of information encoding across multiple modules in our study.

## Implementing multi-module RNNs

The existence of multiple modules in these RNNs was implemented through constraints on the initial recurrent connectivity of the hidden layer. The simplicity (and computational efficiency) of this approach for implementing multi-module RNNs derives from the way that the separation between excitatory and inhibitory units is implemented: all connection weights are passed through a ReLU before they are multiplied by a constant +1/–1 (E vs. I) and applied, so a connection that is culled before training never contributes to the loss, and is never adjusted up or down. To model LIP and PFC, we built networks with two modules, with half of the hidden layer units designated to each module. Each module was allocated half of the excitatory units and half of the inhibitory units in the overall network to ensure that the modules did not differ in their balance of excitation/inhibition prior to training. Motivated by the observation that inhibitory connections in cortex are largely local, we prohibited all inhibitory projections targeting out-of-module neurons. Divisions between brain areas are also distinguished by denser connectivity within areas than between areas, a form of bottleneck that we modeled by restricting the number of excitatory connections between modules—at most 50% of bottom-up connections and 50% of top-down connections. We implemented a weaker form of bottleneck on connections with respect to sensory inputs and motor outputs. No units in module 2 (PFC-like module) could receive projections from the motion-tuned input units, while 50% of units in module 1 (LIP-like module) could receive such projections. Similarly, no units in the LIP-like module could project to the output neurons, while only 50% of the excitatory units in the PFC-like module could. All output drive was restricted to excitatory neurons.

## Analysis of RNN activity

We performed the same analyses on units in the RNNs as we did on the neurophysiology data. As with the neural recordings, we only included the units that showed task-related activity during the test period of the DMC task, defined using the following criteria: (1) the maximum of the averaged activity during the test period should be no less than 0.001; and (2) the activity should exhibit at least one kind of task-related modulation during the test period (such as sample category selectivity, test category selectivity, and M/NM selectivity, one-way ANOVA test, p<0.01).

To compute the latency of M/NM encoding in the LIP and PFC modules in the RNN data, we identified stretches consecutive timesteps where the M/NM signal exceeded its own mean by at least three standard deviations (mean and standard deviation both computed over the first 100 timesteps of the trial, before test onset). The latency was defined as the first timestep of the first such stretch that exceeded 10 timesteps in length.

## Inactivation experiments in silico

To assess the contribution of different RNN units to M/NM decisions, we performed an in silico analogue of neuronal inactivation experiments similar to those used in experimental studies. The monkey experiments revealed a correlation between the activity of PFC nonlinear integrative neurons and the animals' trial-by-trial decisions. As such, we hypothesized that inactivating units with greater nonlinear integrative encoding would have a greater impact on the RNN's ability to perform the task

than inactivating units with weaker nonlinear integrative encoding. To test this hypothesis, we examined the RNNs' behavioral performance after inactivating different subsets of units in the PFC for the duration of the test period during the DMC task (the final 650 ms of each trial). To do this, we first freeze all the parameters of the RNNs after the initial learning. We then divide task-related units in the PFC module into more-nonlinear and less-nonlinear groups, as measured by nonlinearity index. The more- and less-nonlinear groups might project to the output units to different extents, a difference which could explain any divergence in network behavior during inactivation across groups rather than the specific ablation of a local network computation. To control for this possibility, we performed two inactivation experiments: one including all PFC units, the other including only those units in the PFC module that did not directly project to the output units. After selecting the two inactivation groups based on these nonlinearity indices and ensuring that they matched in size, we performed bulk inactivation from the onset of the test stimulus to the end of the trial during DMC. The inactivation was implemented by directly multiplying an activity multiplier ($\leq 1$) to the activity of the target units. In this way, we performed precise, graded causal manipulations; for example, to smoothly titrate the amount of inactivation applied to each unit from complete (100% reduction in activity) to minimal (0% reduction in activity).

We used a similar approach to inactivate feedback projections from the PFC-like module to the LIP-like module. In these projection-specific ablations, rather than multiplying units' activity at each time point by a value between 0 and 1, we multiplied entries of the recurrent weight matrix $W_{rnn}$ by a value ranging between 0 and 1. Multiplication by 0 made LIP entirely immune to the progression of activity in PFC; multiplication by 1 left the interaction between LIP and PFC unaffected relative to what was learned during training. These projection ablations, as with the inactivation of units based on nonlinearity index, were carried out during the test period.

To determine whether, for each network, LIP M/NM encoding was significantly affected by the inactivation of feedback from PFC, we computed at each time point during the test period the correlation between the amount by which projections were multiplied (a vector $v_{true}$ of 10 evenly spaced values between 1,e.g., no inactivation, and 0, e.g., complete inactivation) and the level of M/NM selectivity in LIP at that timestep. If some component of LIP M/NM selectivity is inherited from/arises from PFC feedback, then this correlation between $v_{true}$ and the amount of M/NM selectivity in LIP should differ significantly from baseline. If, however, LIP selectivity for M/NM is entirely independent of PFC feedback, then the amount by which LIP selectivity for M/NM changes should not be consistently related to the amount by which feedback is inactivated. To determine the baseline level of correlation between LIP M/NM selectivity and the amount of PFC feedback inactivation, we computed a full distribution of correlations between $v_{true}$ and each permutation of LIP M/NM selectivities. The number of permutations grows as the factorial of the number of conditions (here, 10 in total), so we limited this analysis to the middle six inactivation conditions/selectivities. To account for the fact that not all permutations are equally 'unrelated' to the true ordering, which influences the level of the correlation obtained, the correlation for each permutation was weighted by the similarity between that permutation and the true ordering. This distribution was then z-transformed and used to compute the p-value of the true ordering's correlation (probability of observing a correlation of equal or greater value among all random permutations of the true ordering). Networks where this p-value fell below 0.001 for a period of at least five consecutive timesteps during the test period were considered to show significant modulation of LIP M/NM selectivity by PFC feedback.

## Acknowledgements

We thank Dr. Pantea Moghimi, Krithika Mohan, and Barbara Peysakhovich for their constructive and helpful comments during the manuscript preparation.

## Additional information

### Funding

| Funder | Grant reference number | Author |
|---|---|---|
| National Institutes of Health | R01EY019041 | David J Freedman |

| U.S. Department of Defense | Vaneevar Bush Faculty Fellowship N000141912001 | David J Freedman |
| National Institutes of Health | T32GM007281 | Ou Zhu |

The funders had no role in study design, data collection and interpretation, or the decision to submit the work for publication.

### Author contributions
Yang Zhou, Conceptualization, Data curation, Formal analysis, Investigation, Methodology, Writing - original draft; Matthew C Rosen, Conceptualization, Formal analysis, Investigation, Methodology, Writing - original draft, Writing - review and editing; Sruthi K Swaminathan, Conceptualization, Data curation, Formal analysis, Investigation, Writing - review and editing; Nicolas Y Masse, Ou Zhu, Investigation, Writing - review and editing; David J Freedman, Conceptualization, Supervision, Funding acquisition, Investigation, Writing - original draft, Project administration, Writing - review and editing

### Author ORCIDs
Yang Zhou ⓘD https://orcid.org/0000-0002-4517-1052
Matthew C Rosen ⓘD https://orcid.org/0000-0002-7868-1702
Nicolas Y Masse ⓘD https://orcid.org/0000-0002-9094-1298
Ou Zhu ⓘD https://orcid.org/0000-0001-8785-8453
David J Freedman ⓘD https://orcid.org/0000-0002-2485-5981

### Ethics
Animal experimentation: This study was performed in strict accordance with the recommendations in the Guide for the Care and Use of Laboratory Animals of the National Institutes of Health. All of the animals were handled according to approved institutional animal care and use committee (IACUC) protocol #71887 of The University of Chicago.

### Decision letter and Author response
Decision letter https://doi.org/10.7554/eLife.58782.sa1
Author response https://doi.org/10.7554/eLife.58782.sa2

## Additional files

### Supplementary files
• Supplementary file 1. Both monkeys' delayed match to category (DMC) task accuracy during match and nonmatch trials.

• Transparent reporting form

### Data availability
Source data has been deposited on FigShare with the following DOI: https://doi.org/10.6084/m9.figshare.13564835.

The following dataset was generated:

| Author(s) | Year | Dataset title | Dataset URL | Database and Identifier |
|---|---|---|---|---|
| Zhou Y, Rosen MC, Swaminathan SK, Masse NY, Freedman DJ | 2021 | Data from: Distributed functions of prefrontal and parietal cortices during sequential categorical decisions | https://doi.org/10.6084/m9.figshare.13564835 | figshare, 10.6084/m9.figshare.13564835 |

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
