## [Decision Letter]

**Acceptance summary:**

This is an important paper for understanding functional contribution of prefrontal (PFC) and parietal areas (LIP and MIP) to sequential decision making. Using a match/non-match (M/NM) decision paradigm in a delayed match-to-category task, the authors focused on the activity during the 250 msec period following the test stimulus presentation. The results reveal that match-preferred PFC neurons exhibited integration of seen and remembered stimuli, indicating that PFC contributes to match/non-match decision process by nonlinear neural integration. Match-preferred LIP neurons contributed primarily to sensory evaluation, while MIP neurons to motor functions such as planning or initiating lever release movements. One of the novelties is the recordings are obtained from three different brain areas from the same monkeys, providing opportunity to assess performance of three areas directly in the same task periods. A second novelty is use of the RNN recurrent neural network model to demonstrate that these outcomes are predicted and to provide a broader context regarding the role of nonlinear units in sequential decision making. The results are broadly consistent with and add to the previous literature.

**Decision letter after peer review:**

Thank you for submitting your article "Distributed functions of prefrontal and parietal cortices during sequential categorical decisions" for consideration by *eLife*. Your article has been reviewed by 2 peer reviewers, and the evaluation has been overseen by a Reviewing Editor and Joshua Gold as the Senior Editor. The following individual involved in review of your submission has agreed to reveal their identity: Klaus Wimmer (Reviewer #1).

The reviewers have discussed the reviews with one another and the Reviewing Editor has drafted this decision to help you prepare a revised submission.

Summary:

This is an interesting paper that addresses the differential roles of PFC, LIP, and MIP in a delayed match-to-category task (selectivity for M vs NM in 3 member categories). This task has been studied extensively in the Freedman lab and several aspects of this dataset have been published previously. Here, the authors focus on the phase of the task from the onset of the second stimulus (test) until the response (button release) approx. 250 ms later. This phase is interesting because it is when the monkeys have to make the decision whether the two stimuli belong to the same or to a different category. The authors find that match-preferred PFC neurons show significantly shorter latency in M/NM selectivity than LIP and MIP and integrate information of both sample and test categories, indicating that PFC contributes to the match/non-match decision process by nonlinear neural integration. Match-preferred LIP neurons were involved in a comparison of sample and test categories, indicating greater role in sensory evaluation, while MIP neurons mainly contribute to motor planning or initiating lever release movements. The results add to the previous literature (e.g. from Romo and Pasternak labs, (Siegel et al. 2015)) and are broadly consistent with those. Overall, this is an interesting topic and recordings in three different brain areas (from the same monkeys) are well suited to address open questions. However, what is new here, as presented, is not sufficiently novel. To bring out the novelty, the concerns, as detailed below, center on improving analyses to better highlight (1) how linear and nonlinear integration in the three brain areas contribute to M/NM decision-making in this task and (2) what kind of functional interactions between these areas leads to a correct M/NM decision. In addition, the use of recurrent neural networks is commended, but (3) needs to go beyond what already has previously been demonstrated; there is opportunity here to provide deeper insight. (4) Substantial re-organization of the manuscript and shortening of introduction and discussion would help improve clarity. We think that these revisions and additional analyses should elucidate the circuits underlying these multi-areal interactions more clearly.

Essential revisions:

Highlighting what is new:

(1) As the differential contribution of information processing of PFC, LIP, MIP has been largely established, the authors must go beyond demonstrating "the relative functions of LIP, PFC, and MIP in sensory, cognitive, and motor functions". Specifically, how do these three brain areas contribute to M/NM decision-making in the DMC task (e.g., linear or non-liner integration, how LIN, NIN, and PN contribute to decision)? and what kind of functional interactions among these brain areas lead to a correct M/NM decision? Please make novelty of this study explicit in abstract, introduction, and discussion.

Data analysis to better isolate roles of linear and non-linear integrating neurons:

(2) Relationship between M/NM effects (match/non-match) and nonlinear integrating neurons (NIN). I was confused to learn first about M/NM neurons (Figure 2) when later – by making a finer distinction – it turns out that many neurons actually respond selectively to only one particular condition (NIN neurons; Figure 7A-C). One key result of Figure 7 is that the nonlinear neurons encode the most information about M/NM. This seems counter-intuitive because a neuron such as the one shown in Figure 7B signals "non-match" only if the test stimulus is from cat 2 and its N/MN effect can't therefore be very high. Some clarifying analysis about the relationship of non-linearity (nonlinearity index, Figure 6B) and match/non-match signals (Figure 2) should be added. At the very least this could be a neuron-by-neuron scatterplot of the non-linearity index vs match/non-match signal in PFC and LIP.

(3) Related with the previous comment, the analyses of mixed selectivity are somewhat convoluted. I would suggest a more standard analysis: as done in (Lindsay et al. 2017), one could model mixed neural responses as a combination of factors (stimuli, decision, etc) and their interaction. Statistics done on the interaction term directly relate to non-linear mixed selectivity would be easier to interpret.

Modelling:

(4) RNNs: The paper shows that the activity in the trained neural network model shares some features with the experimental data but it does not provide a deeper insight into (i) how the M/NM signals are computed in PFC (presumably from NIN neurons in PFC); (ii) whether M/NM is computed in both areas or communicated from the PFC to the LIP module. Moreover, the authors claim that "we developed a novel approach to a multi-module RNN, in which the modularity and rules governing the connections between modules were inspired by neurobiological principles" but the same approach has already been used previously (Kleinman et al. 2019; Song et al. 2016) (Pinto et al. 2019). It has been shown in these previous papers that RNNs can successfully learn tasks under the constraints of a certain network topology. Thus, my question is what we can learn from the RNN here (beyond the proof of principle that the RNN can find a similar solution to the problem)? For example, can the trained RNN be used to generate some non-trivial predictions about the (feed-forward and feedback) interactions between LIP and PFC (see (Kleinman et al. 2019))? Relatedly, no justification is given why the third brain area studied experimentally (MIP) is not included in the model.

(5) Related to the previous comment: It seems that M/NM signals in the model emerge first in LIP (Figure 8C,D), in contrast to the exp. data, where they appear first in PFC. It would also be interesting to see if the model can account for other key features of the data, in particular whether the activity of NIN neurons in the model is predictive of error trials (as shown in Figure 7G for the experimental data).

Manuscript rewriting:

(6) Manuscript is rather descriptive and not well structured (for example, representation of the stimulus category is only addressed in Figure 5, after presenting M/NM selectivity). I think the paper would gain in readability by re-organizing it, by focusing on the key findings, and potentially by moving some of the less central figure panels and control or confirmatory analysis to the supplementary material (e.g. several single neuron examples as in Figure 7; similar results for FEV and decoding e.g. in Figure 4).

(7) Introduction and the discussion are lengthy. In particular, sections of "M/NM selectivity in PFC, LIP, and MIP" and "Comparing the roles of PFC, LIP, and MIP in M/NM decision" are lengthy. We suggest the author shorten and summarize these sections.

[Editors' note: further revisions were suggested prior to acceptance, as described below.]

Thank you for resubmitting your work entitled "Distributed functions of prefrontal and parietal cortices during sequential categorical decisions" for further consideration by *eLife*. Your revised article has been reviewed by 2 peer reviewers and the evaluation has been overseen by Joshua Gold as the Senior Editor, and a Reviewing Editor.

The manuscript has been improved but there are some remaining issues that need to be addressed, as outlined below: Two reviewers, one of whom was a reviewer on your previous submission, have provided feedback. It is clear that substantial work has been done to address the previous set of reviews. Both reviewers do appreciate the RNN approach. Still, they have questions regarding this model. The rationale and meaning of changes to the parameters of the simplified RNN module is not very clear. Without clarification, the choice of parameters appear arbitrary. Further detail is requested to better understand the assumptions and behavior of the model (e.g. how the loss function changes with training; the behavior of firing rates for individual hidden units in each module). Comparison of different modelling choices is requested.

*Reviewer #1:*

The authors have substantially improved the manuscript and they have addressed the issues raised in the first of round of reviews.

I still have some concerns about the RNN modeling approach. I agree that the new modeling results and the analyses of the model are more compelling, and that the model is overall more comparable to the empirical findings presented in the paper. What concerns me is what the authors summarize as follows in their rebuttal letter:

"we dramatically improved and simplified our multi-module RNN modeling approach, including many fewer recurrent units (100 now vs. 400 before) and imposing fewer constraints on connectivity – in particular, eliminating the restriction that units exposed to external inputs/outputs cannot project outside their own module"

What is the biological relevance of a RNN model in which the obtained results depend so dramatically on the number of neurons and on the imposed constraints? Or put another way: what is the justification for using only 10 inhibitory and 40 excitatory units in each of the 2 modules? I understand that these units are not meant to represent single neurons but small populations. The way that the results are presented in the paper suggests that the training was set up with some constraints, then 50 networks were trained and the 41 networks that learnt the task were analyzed. The provided statistics (line 415f) refer to this scenario. But this is not how this research was carried out. First, networks with 200 units were trained and the results were very different, but this is not mentioned anywhere. I think that this can be misleading because it seems that the obtained solution is a robust finding whereas in reality it seems largely to depend on rather arbitrary modeling choices (100 vs 200 units for example). I think that at the very least the old results need to be presented in the paper as well, together with a discussion of the modeling choices and parameters that yield one solution (as in the first submission) or the other solution (as in the second submission).

*Reviewer #3:*

In the current study, Zhou et al., analyzed data based on two previously published paper from the same lab. Although the dataset is not new, the authors have focused on a new question by analyzing choice related signals in the test period, and by comparing among the three areas simultaneously including PFC, LIP and MIP. The authors found that although sensory-categorization signals were strongest and most salient in LIP, the choice related signals are strongest and arose earliest in PFC. By contrast, MIP shows more motor related activity. Importantly, the authors found that there was a group of PFC neurons that nonlinearly integrated signals of remembered sample and visible test stimuli. Finally the authors constructed a RNN with structure based on their neurophysiological findings. Based on this, they were able to show the network hidden units produced similar properties as in the real data. Importantly, by manipulating the units and connections in this network, the authors were able to show the nonlinear PFC-like neurons really play an important causal role in perceptual categorization task. These results are interesting, and particularly with the aid of the RNN, we may get more insights into the neural circuit mechanism mediating perceptual decision making.

1) How is the activity from the remembered stimuli, as well as from the test stimuli extracted from the test 1 period, since the test stimuli were superimposed-shown during that time, and the responses observed during that period should actually be the sum of the two signals? I am not clear about this, and cannot find this clearly in the text.

2) The causal manipulation of RNN demonstrates that the nonlinear PFC is important, but as shown in the neurophysiology, the nonlinear PFC neurons tend to have stronger sensory categorization signals, as they tend to receive stronger connections from LIP neurons. So it is not the "nonlinear" that matters, but the strong sensory categorization signal (potentially from LIP) matters, correct?

3) The description of the RNN is not clear. For example, what is the task of the network (successfully categorize stimuli)? What is its loss function?

4) If all the units in the RNN are excitatory, rather than 10-20% of them are set to be inhibitory, will the model produce similar results and conclusions? The motivation of these settings are not clearly demonstrated in the text.

5) The causal manipulation of RNN demonstrates that the nonlinear PFC is important, but what about LIP? Although overall LIP shows relatively smaller proportion of nonlinear neurons compared to PFC, they still show nonlinear effects. If inactivating these LIP neurons, what would happen? Plus, the authors have emphasized PFC too much by "suppressing" LIP.

---

## [Author Response]

Essential revisions:Highlighting what is new:(1) As the differential contribution of information processing of PFC, LIP, MIP has been largely established, the authors must go beyond demonstrating "the relative functions of LIP, PFC, and MIP in sensory, cognitive, and motor functions". Specifically, how do these three brain areas contribute to M/NM decision-making in the DMC task (e.g., linear or non-liner integration, how LIN, NIN, and PN contribute to decision)? and what kind of functional interactions among these brain areas lead to a correct M/NM decision? Please make novelty of this study explicit in abstract, introduction, and discussion.

We agree with the reviewer’s comment that the original manuscript could have done a better job of clearly describing the novel findings of this study, and the need for additional analysis to go deeper in describing how these areas contribute to sequential decisions. In the revised manuscript, we reorganized the text (including abstract, paragraph 1,2,3,4 in introduction, paragraph 16,17,19,20,22 in the results, and all paragraphs in discussion), added new figures (Figure 7 B and D, Figure 7 Supplement 1 and 3), as well as quantified the correlations between the nonlinear integrative encoding and the encodings of key task variables of neurons in both LIP and PFC. We found significant positive correlations between the degree of nonlinear integrative encoding and the magnitude of task variable encoding (both sample category and M/NM) in PFC. This positive correlation indicates that PFC neurons which showed greater nonlinear integrative encoding were more involved in sample category encoding and M/NM computation, and further emphasizes the preferential role of nonlinear integrative encoding in M/NM decisions.

Specifically, our results suggest that PFC relies on nonlinear integrative encoding of sample and test information to generate M/NM decisions. This is indicated by the observations that (1) M/NM selectivity in PFC emerged with a shorter latency than LIP and MIP, and (2) the neurons which exhibited nonlinear integrative encoding also showed a stronger encoding of the key task variables and were more closely correlated with monkeys’ trial-by-trial M/NM decisions than other neurons in PFC. Our result suggest that LIP is less directly involved in M/NM computation, but is more involved in encoding and maintaining both currently visible and remembered category information during the test period. This is supported by the observation that LIP showed longer latency M/NM selectivity compared to PFC, but stronger and more linear-like integrative sample and test category encoding, compared to PFC. Combining this with the results from our multi-module RNNs, we suggest that the M/NM encoding in LIP is consistent with a top-down signal from higher cortical areas, such as PFC, which appears more directly involved in M/NM computation.

As far as we know, our study is the first attempt to compare how mixed encoding of multiple task variables varies across frontal and parietal cortices in support of decision making. We found that mixed integrative encoding in PFC and LIP may relate to the core functions of each area during the DMC task: In areas closer to sensory input, such as LIP, linear integrative encoding may support independent encoding of visible and remembered stimulus features, and enabling readout of these variables by downstream areas. In areas more associated with cognitive or task-related functions, such as PFC, nonlinear integrative encoding may facilitate the transformation of linear-like encoding observed in upstream areas (e.g. LIP) into more complex nonlinear representations in order to satisfy task demands. Previous studies emphasized an important role of PFC nonlinear mixed encoding for generating higher-dimensional representations which can facilitate downstream linear readout. Extending beyond this, our current study suggests that nonlinear integrative encoding in PFC is a key mechanism for manipulating the encoding of sensory stimuli and items in working memory to form decision related (M/NM) representations.

Furthermore, we trained multi-module RNNs to perform the DMC task and analyzed neural activity and the connectivity within and between each module during task performance. The units of our two-module RNNs exhibited patterns of activity highly similar to those observed in the neurophysiology data, including both the key features of information encoding within LIP/PFC and the functional differences between LIP and PFC. To further understand the circuit mechanisms of M/NM computation and the roles of nonlinear integration of task variables during M/NM decisions, we implemented causal perturbations of the RNNs’ connectivity and activity. These included two forms of causal perturbation – targeted silencing of neurons based on the extent to which they show a particular pattern of information encoding (in this case, nonlinear integration of sample and test stimuli), and ablating connections based on the identities of the pre- and post-synaptic modules. Using this type of functional targeting, we grained insight into the mechanisms underlying M/NM computations, as well as the importance of nonlinear mixed encoding during sequential decisions. First, we examined the functional interaction between LIP and PFC modules, and found that M/NM selectivity in the LIP module was significantly modulated by top-down input from PFC module. Second, we validated the necessity of nonlinear integrative encoding for mediating M/NM decisions during the DMC task. These are important complements to our experimental results which demonstrate correlations between the activity of different neurons in PFC/LIP with monkeys’ trial-by-trial decisions. We also examined the connectivity both within and between the RNNs’ different modules. We found that the nonlinear integrative units in the PFC-like module were more likely to be ‘hub neurons’ with broad connections within the network, mediating the integration and transformation of information crucial for solving the task. These forms of correlational and causal evidence, collected within the same model system, highlight the role that RNNs can play in (a) revealing the circuit basis of cognitive computations, and (b) motivating future experiments to test model predictions.

In order to more clearly describe these points, we have reorganized the abstract, introduction, and Discussion sections extensively to cover the novel points of this study more explicitly. Specifically, we shortened the introduction, primarily focusing on the mechanisms underlying the M/NM computation and PFC nonlinear integrative encoding. We also substantially rewrote the discussion to more concisely discuss the main novel findings of the study and their implications.

Data analysis to better isolate roles of linear and non-linear integrating neurons:(2) Relationship between M/NM effects (match/non-match) and nonlinear integrating neurons (NIN). I was confused to learn first about M/NM neurons (Figure 2) when later – by making a finer distinction – it turns out that many neurons actually respond selectively to only one particular condition (NIN neurons; Figure 7A-C). One key result of Figure 7 is that the nonlinear neurons encode the most information about M/NM. This seems counter-intuitive because a neuron such as the one shown in Figure 7B signals "non-match" only if the test stimulus is from cat 2 and its N/MN effect can't therefore be very high. Some clarifying analysis about the relationship of non-linearity (nonlinearity index, Figure 6B) and match/non-match signals (Figure 2) should be added. At the very least this could be a neuron-by-neuron scatterplot of the non-linearity index vs match/non-match signal in PFC and LIP.

We agree that it seems counter-intuitive that the NINs showed significantly greater and shorter-latency M/NM selectivity than the pure-M/NM selective neurons in PFC, as the NINs generally encoded a subset of the four sample-test combinations. Intuitively, it seems that the pure M/NM selective neurons should show greater M/NM selectivity than the NINs. However, the population of pure-M/NM-selective neurons in PFC did not show as strong M/NM encoding compared to areas more associated with motor output such as MIP. To better clarify this in the revised manuscript, we have followed the reviewers’ suggestions and added neuron-by-neuron scatterplots of the nonlinearity index vs M/NM selectivity in PFC (Figure 7B and D) and LIP (figure 7 supplement 3), and calculated the correlation between the degree of nonlinear integrative encoding (shown by the nonlinearity index) and the M/NM selectivity of PFC neurons (both magnitude and latency). We found significant positive correlations between the degree of nonlinear integrative encoding and the magnitudes of M/NM encoding (r = 0.75, p = 2.0×10^-27^). Such positive correlations indicate that PFC neurons which showed greater levels of nonlinear integrative encoding were more likely to be involved in M/NM computation. We also found a significant negative correlation between the degree of nonlinear integrative encoding and the latency of M/NM selectivity in PFC (r = -0.44, p = 2.3×10^-5^), suggesting that the PFC neurons which showed greater nonlinear encoding tended to encode M/NM with a shorter latency. These two analyses further support the conclusion that NINs were more involved in M/NM computation than the pure-M/NM-selective neurons in PFC, and that PFC nonlinear integrative encoding is suited to facilitate sequential M/NM decisions.

(3) Related with the previous comment, the analyses of mixed selectivity are somewhat convoluted. I would suggest a more standard analysis: as done in (Lindsay et al. 2017), one could model mixed neural responses as a combination of factors (stimuli, decision, etc) and their interaction. Statistics done on the interaction term directly relate to non-linear mixed selectivity would be easier to interpret.

We realize that the original manuscript was unclear in describing our analysis approach. We actually used the same analysis as in the study mentioned by the reviewer (i.e. two-way ANOVA). We defined the nonlinear integrative neurons as those that exhibited significant sample and test category selectivity, as well as significant interaction between sample and test category selectivity. Meanwhile, we identified the neurons which exhibited a significant interaction between sample and test category selectivity, but non-significant sample and test category selectivity, as pure M/NM selective neurons. We did not classify the pure M/NM selective neurons as mixed selective because they exhibited neither significant sample category selectivity nor significant test category selectivity. Therefore, the interaction term cannot be directly related to the nonlinear mixed selectivity in our study. In order to diminish the influence of pure M/NM selectivity on the calculation of nonlinearity index, we used the following formula to quantify the level of nonlinear integrative encoding: NLI = | FEV _(S1T1 vs. S1T2)_ – FEV _(S2T1 vs. S2T2)_| (please see detail in Methods). In this analysis, if a neuron showed only pure M/NM selectivity, the value of its nonlinearity index will be 0. In the revised manuscript, we substantially modified the results and methods sections to more clearly describe this method used for classifying neuron types, and cited the paper mentioned by the reviewer. We also added a supplementary figure (Figure 7 Supplement 1) to show the results of the two-way ANOVA in both LIP and PFC, and to clarify the approach used for classifying different types of neurons as well as the results.

Modelling:(4) RNNs: The paper shows that the activity in the trained neural network model shares some features with the experimental data but it does not provide a deeper insight into (i) how the M/NM signals are computed in PFC (presumably from NIN neurons in PFC); (ii) whether M/NM is computed in both areas or communicated from the PFC to the LIP module. Moreover, the authors claim that "we developed a novel approach to a multi-module RNN, in which the modularity and rules governing the connections between modules were inspired by neurobiological principles" but the same approach has already been used previously (Kleinman et al. 2019; Song et al. 2016) (Pinto et al. 2019). It has been shown in these previous papers that RNNs can successfully learn tasks under the constraints of a certain network topology. Thus, my question is what we can learn from the RNN here (beyond the proof of principle that the RNN can find a similar solution to the problem)? For example, can the trained RNN be used to generate some non-trivial predictions about the (feed-forward and feedback) interactions between LIP and PFC (see (Kleinman et al. 2019))? Relatedly, no justification is given why the third brain area studied experimentally (MIP) is not included in the model.

We agree with the reviewer’s comment that the original RNN modeling and analysis could have done a better job of providing insight into the circuit mechanisms underlying M/NM computation. In the revised manuscript, we dramatically improved and simplified our multi-module RNN modeling approach, including many fewer recurrent units (100 now vs. 400 before) and imposing fewer constraints on connectivity – in particular, eliminating the restriction that units exposed to external inputs/outputs cannot project outside their own module. We also added new analyses to examine both the feedforward and feedback connection weights across modules. This allowed a more detailed description of the circuit mechanisms by which PFC and LIP modules participate in M/NM computation in the RNNs. Specifically, this revealed three new results in the revised RNNs: 1) M/NM selectivity in the PFC module emerged with shorter or similar latency compared to the LIP module, which is in closer agreement with the experimental data compared that described in the original version of the manuscript; 2) M/NM selectivity in the PFC module is not inherited from LIP; 3) M/NM selectivity of LIP units is significantly modulated by feedback input from the PFC module. These results suggest that PFC module was likely to be the source for M/NM computation, and that M/NM selectivity in the LIP module likely reflects feedback from the PFC module.

We also performed new analyses of the model to investigate the circuit mechanisms that underlie the nonlinear integration of sample and test category information to form the M/NM decision. In particular, we identified three key features: (1) PFC module neurons that more nonlinearly integrate sample/test information are more likely to project to decision outputs, suggesting that nonlinear integration may play a proximal role in decision execution; (2) PFC module units showing greater nonlinear integration receive greater feedforward input from the LIP module and are more recurrently connected with other PFC-module units, compared to the PFC-module units with less nonlinear integrative encoding, suggesting that PFC nonlinear integrative neurons might act as ‘hubs’ mediating information integration; and (3) causal evidence, obtained by inactivating PFC modules’ most and least nonlinear units that do vs. do not project to decision outputs, shows that nonlinear integration plays a key role in decision formation; Combining these results, the multi-module RNNs provide deeper insights (compared to the original manuscript) into the potential circuit mechanisms and functional interactions between modules which underlie their M/NM decisions during the DMC task, as well as the importance of nonlinear task variable encoding during sequential decisions.

Furthermore, we agree with the reviewer that it is not proper to claim ‘a novel approach to a multi-module RNN’, since recent studies have also begun to use constraints on initial weights to train multi-module RNNs on cognitive tasks. To reflect this, we no longer refer to our approach for generating multi-module RNNs as ‘novel’, although we do maintain that several aspects of our results are novel. We do want to emphasize the novelty of the causal experiments we use multi-module RNNs to perform. In our view, one of the key advantages of RNNs as a model system is the relative ease with which they permit causal perturbations analogous to those used in experiments – in particular, temporally-precise, graded, targeted inactivations. When performed in vivo, these inactivations have yielded evidence indispensable for generating, testing, and refining hypotheses about neural circuit mechanisms. However, the difficulty of executing these perturbations in primate systems is prohibitive, particularly when trying to assess the effect of inactivating specific types of neurons in one area while simultaneously examining the activity in another area. Given these difficulties, we feel that demonstrating a similar approach to generate hypotheses about the interactions between brain areas in silico represents an important advance. We specifically discussed the benefits of our modelling approach in the discussion as following:

‘These RNN models’ primary benefits are twofold: first, they offer full knowledge of network connectivity; and second, they facilitate causal perturbations analogous to those used in experimental preparations. […] These are important complements to our experimental results, which demonstrate correlations between the activity of different neurons in PFC/LIP with monkeys’ trial-by-trial decisions, and make predictions which can guide future experiments and data analysis.’.

Lastly, our neurophysiological results showed that MIP activity primarily reflected the monkeys’ arm and/or hand movements used to report their decision, but did not appear to play a direct role in M/NM decisions in the DMC task. In this study, we implemented multi-module RNNs to further examine the circuit mechanisms underlying the M/NM decision, but not specifically the actions used to report those decisions. Therefore, we did not include a third module to simulate a more motor pool of neurons (corresponding to motor networks including MIP) within our RNNs. In the revised manuscript, we added the following sentence in the Results section to clarify this point:

‘Because our neurophysiological results suggest that MIP did not play a direct role in M/NM decisions in the DMC task, we designed and implemented RNNs composed of two hierarchically organized modules, with the module closer to sensory input intended to correspond to LIP, and the module closer to the behavioral output corresponding to PFC..’

We also agree with the reviewer that including more realistic motor output in future RNNs could be very useful for understanding how decisions are transformed into specific actions. Therefore, we added the following sentence in the discussion:

‘Future studies with RNNs that include more realistic motor output modules will likely be useful for understanding how decisions are transformed into specific actions.’.

(5) Related to the previous comment: It seems that M/NM signals in the model emerge first in LIP (Figure 8C,D), in contrast to the exp. data, where they appear first in PFC. It would also be interesting to see if the model can account for other key features of the data, in particular whether the activity of NIN neurons in the model is predictive of error trials (as shown in Figure 7G for the experimental data).

We thank the reviewer for highlighting this important point, as it directly inspired the revision of our modeling approach in the revised manuscript. In trying to understand why the previous models failed to recapitulate the M/NM latency difference seen in the data, we discovered that the increased prevalence of M/NM encoding in the PFC module was due to selective transmission of M/NM information that had already been computed in LIP. LIP units that projected to PFC were especially selective for M/NM, and units that did not directly project to PFC tended to show low M/NM selectivity. Because the LIP module but not the PFC module was driven directly by external sensory inputs, LIP continued to encode the test stimulus strongly for the duration of the test period, but selectively transmitted M/NM information downstream. We hypothesized that this signaled the settling of networks on a pathological set of solutions, where the LIP module was sufficiently large to ‘claim’ all of the main elements of the decision computation—generation of the categorical representation, maintenance of sample information during the delay, and the eventual computation of M/NM. To address this, we reduced network size, including 50 rather than 200 units in each module. Because we had increased network sizes initially to promote network trainability (small networks with many constraints on initial connectivity tended to converge only very slowly during training), we relaxed several of our original constraints on connectivity. In particular, we allowed connections to PFC from excitatory LIP units driven directly by sensory inputs, as well as connections to LIP from excitatory PFC units that directly drive outputs. This yielded networks where M/NM latency was either similar in LIP vs. PFC or shorter in PFC, consistent with the experimental data. With regard to the error trials, we strongly agree with the reviewer in the importance of showing that the NIN neurons correlate more closely with networks’ trial-by-trial responses. However, our training method resulted in converged networks with very high accuracy (>99%), which prohibits a robust error trial analysis as with the experimental data. During the process of revisions, we experimented with a training method to try to generate networks with more variable performance (training networks on single examples at a time rather than on mini-batches of tens to hundreds of trials). However, this training was prohibitively time-consuming and much more inconsistent than our standard training protocol, issues that we hope to ameliorate in future work to facilitate error-trial analyses. Again, we thank the reviewers for their insightful point about the latency difference in M/NM encoding between modules differing with respect to the experimental data, as it illuminated a pathological aspect of our initial RNNs’ solution, and a major improvement in the revised modeling work. We feel that addressing this shortcoming has substantially improved our study.

Manuscript rewriting:(6) Manuscript is rather descriptive and not well structured (for example, representation of the stimulus category is only addressed in Figure 5, after presenting M/NM selectivity). I think the paper would gain in readability by re-organizing it, by focusing on the key findings, and potentially by moving some of the less central figure panels and control or confirmatory analysis to the supplementary material (e.g. several single neuron examples as in Figure 7; similar results for FEV and decoding e.g. in Figure 4).(7) Introduction and the discussion are lengthy. In particular, sections of "M/NM selectivity in PFC, LIP, and MIP" and "Comparing the roles of PFC, LIP, and MIP in M/NM decision" are lengthy. We suggest the author shorten and summarize these sections.

We have followed the reviewers’ suggestions in order to improve the organization and presentation of the manuscript. We shortened the introduction and Discussion sections, and rewrote much of the Discussion section to make it clearer and more concise. We also reorganized the Results section in the revised manuscript. We also moved the original figure G-H, figure 7A-C to supplementary figures as suggested by the reviewer.

[Editors' note: further revisions were suggested prior to acceptance, as described below.]

The manuscript has been improved but there are some remaining issues that need to be addressed, as outlined below: Two reviewers, one of whom was a reviewer on your previous submission, have provided feedback. It is clear that substantial work has been done to address the previous set of reviews. Both reviewers do appreciate the RNN approach. Still, they have questions regarding this model. The rationale and meaning of changes to the parameters of the simplified RNN module is not very clear. Without clarification, the choice of parameters appear arbitrary. Further detail is requested to better understand the assumptions and behavior of the model (e.g. how the loss function changes with training; the behavior of firing rates for individual hidden units in each module). Comparison of different modelling choices is requested.Reviewer #1:The authors have substantially improved the manuscript and they have addressed the issues raised in the first of round of reviews.I still have some concerns about the RNN modeling approach. I agree that the new modeling results and the analyses of the model are more compelling, and that the model is overall more comparable to the empirical findings presented in the paper. What concerns me is what the authors summarize as follows in their rebuttal letter:"we dramatically improved and simplified our multi-module RNN modeling approach, including many fewer recurrent units (100 now vs. 400 before) and imposing fewer constraints on connectivity – in particular, eliminating the restriction that units exposed to external inputs/outputs cannot project outside their own module"What is the biological relevance of a RNN model in which the obtained results depend so dramatically on the number of neurons and on the imposed constraints? Or put another way: what is the justification for using only 10 inhibitory and 40 excitatory units in each of the 2 modules? I understand that these units are not meant to represent single neurons but small populations. The way that the results are presented in the paper suggests that the training was set up with some constraints, then 50 networks were trained and the 41 networks that learnt the task were analyzed. The provided statistics (line 415f) refer to this scenario. But this is not how this research was carried out. First, networks with 200 units were trained and the results were very different, but this is not mentioned anywhere. I think that this can be misleading because it seems that the obtained solution is a robust finding whereas in reality it seems largely to depend on rather arbitrary modeling choices (100 vs 200 units for example). I think that at the very least the old results need to be presented in the paper as well, together with a discussion of the modeling choices and parameters that yield one solution (as in the first submission) or the other solution (as in the second submission).

We thank the reviewer for highlighting this ambiguity, and have realized that we did not sufficiently explain the full rationale behind the changes we made. The revised manuscript includes a new paragraph in the discussion which describes the range of parameters and constraints which we tried on our path to developing the RNN models used in the current version of the manuscript, as well as additions to the results and methods sections to address the same point.

The current revision of the manuscript focused on clarifying the RNN modeling approach, in particular (a) the rationale behind the changes made to the models in the previous revision, and (b) the approach’s general assumptions and their effects on the models’ behavior. Below we discuss these points in detail, noting in particular where/how we address them in the new version of the manuscript. The reviewer is correct that the description of the modeling approach and results in the previous version of the manuscript did not describe the sequence of events and findings that led to the choices of model architecture and parameters. That sequence can be summarized as follows:

(a) We developed an approach for training multi-module RNNs to perform cognitive tasks, like DMC, which included several constraints (past those introduced by existing approaches, e.g. that of Kleinman et al. 2019) intended to differentiate the computations performed by each module. Training networks with these constraints was more difficult than training networks without, and as a result, we used relatively large networks (200 units per module, for a total of 400 units in the network) to increase the probability of our networks converging to a high level of task performance.

(b) We analyzed these multi-module RNN models using the same approaches used on the neural recordings, and observed some notable similarities with features of information encoding in the recorded data.

(c) These models failed to reproduce one of the key features seen in the data – the tendency of M/NM selectivity to manifest earlier in PFC than in LIP.

We identified two candidate explanations for this discrepancy: (a) the earlier emergence of the M/NM in the LIP-like module was incidental (e.g. not critical for task performance), consistent with our conclusion based on neural data; (b) the original models were adopting a pathological solution to the task – pathological in the sense that networks were effectively solving the task in one module or the other, despite the architectural constraints designed to promote modular solutions. This possibility stems from the fact that we encourage networks to be modular by endowing their structure with modularity, but cannot directly require these structural modules to also be strongly modular in function. If the first possibility were true (earlier M/NM encoding in LIP module being incidental), then the units projecting from the LIP module to the PFC module should predominantly convey a sample/test category-selective signal, and not a strong M/NM signal. We found the opposite: the units projecting forward from the LIP module conveyed a strongly M/NM selective signal, and in most networks the tendency of a unit to project forward from LIP corresponded with its M/NM selectivity. This led us to focus on the other possibility – that the original networks had adopted a pathological solution to the task, likely due to at least a subset of the constraints we imposed on connectivity/communication between modules. Specifically, the evidence suggested to us that the task was being solved primarily through activity in the LIP module, with the PFC module serving largely to mirror this information to drive the network’s behavioral output units. In other words, the task solution really was not taking advantage of the potential to divide task-related computations across the two modules. This interpretation is wholly consistent with (a) the tendency of the PFC module to be overwhelmingly selective for M/NM rather than sample or test categories during the test period, and (b) the consistently longer-latency representation of M/NM in PFC vs. in LIP.

In light of this alternate explanation – that our original constraints on connectivity required large networks to stably converge, and that solutions obtained by these large networks did not effectively modularize – we rethought our approach to generating multi-module RNNs. Specifically, we eliminated several of the constraints on connectivity between modules which had obstructed network convergence, and had initially pushed us to use large networks. The primary alternation was thus a relaxation of the restriction that units directly receiving external/sensory inputs and units directly projecting to behavioral outputs cannot themselves form synaptic connections with one another. We initially enforced this restriction to encourage the processing of sensory inputs via recurrent connections in the LIP module. However, without specific connectomic data to support its inclusion, and because it increased the difficulty of network training prohibitively, we chose to eliminate this constraint. Our original networks were also trained to perform multiple tasks to encourage the computation of category signals via recurrent connections rather than through specific plasticity of the input projections. However, to better match the experimental setting, where animals were primarily trained to perform the DMC task, here we opted instead to freeze the input weights, ensuring that plasticity related to category computation would occur inside the recurrent connections of the network rather than at the level of the inputs. The net effect of these changes was a reduction in the number of constraints required to generate multi-module networks, and thus a simpler approach overall. We confirmed that this approach succeeded in generating networks that would reliably converge (>= 80% of the time), even at much smaller sizes (100 neurons total, so 50 neurons per module). To speed up downstream analyses of these networks, where runtime grows in proportion with network size, we chose to train and analyze networks with 100 neurons.

In order to verify that our RNN modeling results did not depend on the number of neurons, we trained more example networks with different numbers of neurons (100, 200, 300, 400), using the same constraints for modularity as the networks presented in the current version of the manuscript. We added the results of these example networks as a new supplementary figure in the revised manuscript (Figure 8 Supplementary 1). The key results are consistent between the networks with different sizes: 1) the LIP module encoded both the sample and test category more strongly than the PFC module; 2) the PFC module encoded M/NM more strongly than the LIP module; 3) M/NM selectivity in the PFC module arises with a latency similar to or shorter than in the LIP module, which is consistent with both the experimental data and the network results in the main manuscript; 4) the PFC module showed greater nonlinear integrative encoding than LIP module. These results confirmed that the results we obtained are robust to network size within a reasonable range, and thus that network size was not the key constraint for determining the characteristics of information encoding across multiple modules in our study.

In order to clarify this point, we added the following text in the revised Results section:

‘Because we found similar results using networks across a range of sizes (n = [100,200,300,400}; Figure 8, Supplementary 1), all results discussed henceforth used networks with n=100 hidden units. We independently trained 50 such networks with randomly initialized weights and identical hyperparameters to perform the DMC task, using methods previously described (Masse et al., 2019).’

We also added the following text in the revised Methods section:

‘In order to verify that our RNN modeling results did not critically depend on the number of neurons, we trained more example networks with different numbers of neurons (100, 200, 300, 400), using the same constraints for modularity (Figure 8 Supplementary 1). We found that the key results are consistent between the networks with different sizes, confirming that the results we obtained are robust to network size within a reasonable range, and thus that network size was not the key constraint for determining the characteristics of information encoding across multiple modules in our study.’

Furthermore, we also added several sentences in the Results section, as well as one paragraph in the Discussion section to discuss the choice of connectivity constrains for generating modularity:

Results: “…In designing our multi-module modelling approach, we also tested several other methods of defining RNN modules, e.g. via additional constraints on the identity/number of projections across modules. However, we chose the constraints used here because they consistently yielded networks whose structurally-defined modules also manifested different patterns of activity and task-variable encoding, suggesting effectively modular solutions to the task…”

Discussion: ‘Critically, however, we note that not all approaches to enforcing modularity by constraining connectivity result in networks whose solutions are also modular in function. […] Although we also explored additional restrictions on how modules communicate, including the prohibition of direct connections between units receiving external sensory inputs and units projecting to the network’s behavioral output, these additional restrictions did not help to yield networks matching the key features of the neural data in the current study.’.

Since we reproduced the characteristic latency difference between M/NM encoding in PFC vs. LIP in the data, as well as the other key features observed in our neural data, across the networks within a reasonable size range, we feel that it might not be necessary to include the initial RNN modelling results in the manuscript. However, we have added additional description of the range of modeling constraints and parameters explored before settling on the final model presented in the current manuscript (see above). Nevertheless, if the reviewer think it is necessary to show the initial networks in the current manuscript, we would also be happy to include these results.

Reviewer #3:[…] 1) How is the activity from the remembered stimuli, as well as from the test stimuli extracted from the test 1 period, since the test stimuli were superimposed-shown during that time, and the responses observed during that period should actually be the sum of the two signals? I am not clear about this, and cannot find this clearly in the text.

We realize that our description of this analysis in the original manuscript requires additional explanation. As pointed out by the reviewer, the test period activity reflected both sample and test category information. In Figure 5, we showed that test period activity in PFC and LIP, but not MIP, significantly encoded both sample and test category information. Some neurons significantly showed both sample and test category selectivity, and some neurons only significantly encoded either sample or test category information. We performed a two-way ANOVA on test-period activity with sample and test categories as factors to quantify the selectivity profile of neurons in all three cortical areas. The magnitude of the category selectivity was quantified using a fraction explained variance analysis. In the revised manuscript, we revised the text in the Results section as follows to clearly describe this analysis:

‘We first quantified the neuronal representation of sample and test categories in each area during the first test period using a two-way ANOVA on test-period activity with sample and test categories as factors. The magnitude of the category selectivity was quantified using unbiased fraction explained variance (see methods).’

2) The causal manipulation of RNN demonstrates that the nonlinear PFC is important, but as shown in the neurophysiology, the nonlinear PFC neurons tend to have stronger sensory categorization signals, as they tend to receive stronger connections from LIP neurons. So it is not the "nonlinear" that matters, but the strong sensory categorization signal (potentially from LIP) matters, correct?

We thank the reviewer very much for raising this question. Indeed, it not clear whether the deficit in accuracy that results from inactivating the nonlinear PFC units reflects (a) the elimination of strong category signal inherited from the LIP module, or (b) inhibition or ablation of the nonlinear format of information integration. In the current study, we found that a group of PFC neurons which exhibited nonlinear integrative encoding played a more proximal role in the M/NM decisions, compared to the other PFC neurons. This is supported by the neural evidence that PFC nonlinear units were more involved in the encoding of key task factors (stronger category encoding, stronger and shorter-latency M/NM encoding) and more correlated with monkeys’ trial-by-trial decisions. In order to explore the potential circuit mechanisms enabling nonlinear integration during M/NM decisions, we examined the connection weights within our multi-module RNNs. We found that the PFC nonlinear units were more likely to be connection ‘hubs’ within the PFC module within our RNNs. In addition to receiving stronger input from the LIP module, nonlinear PFC units were more recurrently connected within the PFC module, and projected more heavily to the output units, compared to the other PFC units. We think all three of these characteristics may contribute to the role of nonlinear PFC units in the M/NM decisions. To directly test this, we analyzed the neural selectivity of PFC units for sample category, test category, and M/NM after complete inactivation of the linear or nonlinear groups of PFC units. Consistent with the interpretation that nonlinear neurons’ role reflects (at least in part) their tendency to strongly represent stimulus category (both sample and test), networks’ averaged category selectivity (both sample and test) decreased much more severely after inactivating the nonlinear group than the linear group. (We view this as in line with the reviewer’s suggestion that at least part of the deficit in accuracy when inactivating nonlinear neurons reflects a quenching of category signals). Further, we observe that the M/NM selectivity of PFC units decreases more strongly after inactivating the nonlinear vs. the linear group of PFC neurons. Interestingly, this disruption was even greater for the M/NM selectivity than the category selectivity, suggesting that at least some of the deficit observed reflects the integration of sample/test category information in particular.

Therefore, we agree with the reviewer that the strong category signal (potentially resulting from the strong feed forward projections from LIP that nonlinear PFC units receive) is a critical component in the central role played by nonlinear PFC neurons in M/NM decisions. However, we think it might not be the only key factor, as the strong and short-latency M/NM signal is also likely to play an important role.

We believe that this question which the reviewer has raised – whether the format of information encoding/integration or specific forms of network connectivity within the PFC played a dominant role in the M/NM computation – is an extremely important one. However, the gold standard of evidence for answering it would require additional neuronal measurements and causal experiments which go beyond the scope of this study. Because most of the recordings we analyzed were collected in separate sessions, here we cannot resolve connections between LIP and PFC neurons. Unfortunately, it is no longer possible to conduct additional recordings in the animals from the original study reported here. However, in our future work, we hope to use high-throughput, multichannel recordings as well as casual approaches (such as inactivation or stimulation experiment) to examine more directly the relationship between connectivity and information encoding/integration of information across different brain areas.

3) The description of the RNN is not clear. For example, what is the task of the network (successfully categorize stimuli)? What is its loss function?4) If all the units in the RNN are excitatory, rather than 10-20% of them are set to be inhibitory, will the model produce similar results and conclusions? The motivation of these settings are not clearly demonstrated in the text.

We agree with the reviewers that previous versions of the manuscript did not make sufficiently clear the general assumptions of this modelling approach, nor how these assumptions/architecture choices affect model behavior. To address this, we have added explanatory material to both (a) the main text, in the portion of the Results where we discuss the models, and (b) the Methods section. In the revised manuscript (Results and Methods sections), we now clarify basic details of the modelling setting, e.g. what loss function is used, motivation for the architectural choices adopted, the general trend of neuronal activities throughout the task, etc. (Results section, “PFC NINs are crucial for solving the M/NM computation in silico”):

‘The task design of the DMC task the models were trained to perform was tailored to match those used in experiments: the task sequence and motion directions were the same as in the monkey experiments, and the networks were trained to indicate whether the sample and test stimuli belonged to the same category. Network parameters (recurrent weights/biases and output weights) were optimized to minimize a loss function with three parts: 1) one related to performance of the DMC task (cross-entropy of the network’s generated outputs with respect to the correct outputs), 2) a metabolic cost on firing rates, and 3) a metabolic cost on connectivity (see details in Methods).’

We expound on these details further in the Methods section (“Recurrent neural network training”). In particular, we explain in more detail the logic underlying the parameter choices that we use (e.g. E/I ratio), and emphasize their relation to standard choices made among previously-published studies that adopt a related modelling approach:

“… The E/I ratio (80% excitatory to 20% inhibitory) was chosen to be consistent with the range of proportions found in the mammalian neocortex (Markram et al. 2004), and accordingly reflects the standard among models constrained to obey Dale’s principle. […] Trials were programmatically generated by choosing sample/test stimulus identities and using these to construct inputs to the networks at each timestep (based on the motion tuning of the input layer) and desired output (fixation, match, non-match) at each timestep.”

5) The causal manipulation of RNN demonstrates that the nonlinear PFC is important, but what about LIP? Although overall LIP shows relatively smaller proportion of nonlinear neurons compared to PFC, they still show nonlinear effects. If inactivating these LIP neurons, what would happen? Plus, the authors have emphasized PFC too much by "suppressing" LIP.

We thank the reviewer for this question – we have wondered similarly about the role of LIP nonlinear neurons, and in fact performed versions of the inactivation experiment that the reviewer has asked us to speculate about. In all cases, we found that suppressing or extinguishing LIP neurons’ activity catastrophically impairs networks’ ability to generate correct M/NM decisions. However, it is difficult to dissociate how much of this effect is due to the disruption of nonlinear integration occurring at the level of single neurons in LIP, as opposed to a general lesion of feedforward stimulus drive. This is compounded by the fact that all sensory inputs that arrive at the PFC module flow originally through the LIP module in our networks, which rules out the possibility of conducting bulk LIP inactivations without disrupting stimulus input to the PFC. Finally, we found the models’ PFC modules to consist of a ‘continuum of nonlinearity’, ranging from some units that linearly integrated information to neurons that integrated information in a highly nonlinear way. This was not the case in the LIP module, where most of the units displayed low indices of nonlinear integration. Given these three factors – the difficulties of interpreting LIP inactivation assays in silico; the overall higher level of nonlinear encoding displayed by the PFC, both in the models and the data; and the presence of distinct neuron groups in the PFC that integrate stimuli in a substantially linear vs. non-linear way – we prioritized the PFC as the site of inactivations targeted to the more vs. less nonlinear groups of neurons. On the other hand, our neural data showed that the M/NM encoding arises with a shorter latency in PFC than in LIP. This indicates that PFC rather than LIP is likely to be the brain area which leads the M/NM decision process. In addition, our RNN modelling results also showed that the M/NM selectivity in the LIP module was at least partially inherited from the top-down input from PFC, suggesting a minor role for LIP in integrating the sample and test information.